# ARE MORE LAYERS BENEFICIAL TO GRAPH TRANSFORMERS?

**Haiteng Zhao[1]\*, Shuming Ma[2], Dongdong Zhang[2], Zhi-Hong Deng[1]†, Furu Wei[2]**
[1] Peking University
[2] Microsoft Research
`{zhaohaiteng,zhdeng}@pku.edu.cn`
`{shumma,dozhang,fuwei}@microsoft.com`

## ABSTRACT

Despite that going deep has proven successful in many neural architectures, the existing graph transformers are relatively shallow. In this work, we explore whether more layers are beneficial to graph transformers, and find that current graph transformers suffer from the bottleneck of improving performance by increasing depth. Our further analysis reveals the reason is that deep graph transformers are limited by the vanishing capacity of global attention, restricting the graph transformer from focusing on the critical substructure and obtaining expressive features. To this end, we propose a novel graph transformer model named DeepGraph that explicitly employs substructure tokens in the encoded representation, and applies local attention on related nodes to obtain substructure based attention encoding. Our model enhances the ability of the global attention to focus on substructures and promotes the expressiveness of the representations, addressing the limitation of self-attention as the graph transformer deepens. Experiments show that our method unblocks the depth limitation of graph transformers and results in state-of-the-art performance across various graph benchmarks with deeper models.

## 1 INTRODUCTION

Transformers have recently gained rapid attention in modeling graph-structured data (Zhang et al., 2020; Dwivedi & Bresson, 2020; Maziarka et al., 2020; Ying et al., 2021; Chen et al., 2022). Compared to graph neural networks, graph transformer implies global attention mechanism to enable information passing between all nodes, which is advantageous to learn long-range dependency of the graph stuctures (Alon & Yahav, 2020). In transformer, the graph structure information can be encoded into node feature (Kreuzer et al., 2021) or attentions (Ying et al., 2021) by a variant of methods flexibly with strong expressiveness, avoiding the inherent limitations of encoding paradigms that pass the information along graph edges. Global attention (Bahdanau et al., 2015) also enables explicit focus on essential parts among the nodes to model crucial substructures in the graph.

Graph transformer in current studies is usually shallow, i.e., less than 12 layers. Scaling depth is proven to increase the capacity of neural networks exponentially (Poole et al., 2016), and empirically improve transformer performance in natural language processing (Liu et al., 2020a; Bachlechner et al., 2021) and computer vision (Zhou et al., 2021). Graph neural networks also benefit from more depth when properly designed (Chen et al., 2020a; Liu et al., 2020b; Li et al., 2021). However, it is still not clear whether the capability of graph transformers in graph tasks can be strengthened by increasing model depth. So we conduct experiments and find that current graph transformers encounter the bottleneck of improving performance by increasing depth. The further deepening will hurt performance when model exceeds 12 layers, which seems to be the upper limit of the current graph transformer depth, as Figure 1 (left) shows.

In this work, we aim to answer why more self-attention layers become a disadvantage for graph transformers, and how to address these issues with the proper model design. Self-attention (Bahdanau et al., 2015) makes a leap in model capacity by dynamically concentrating on critical parts

---

\*Work done during internship at Microsoft
†Corresponding Author

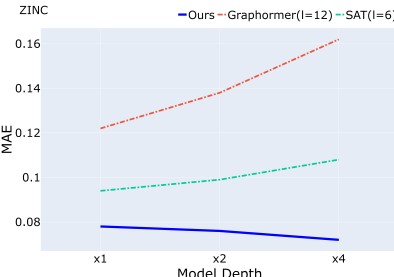 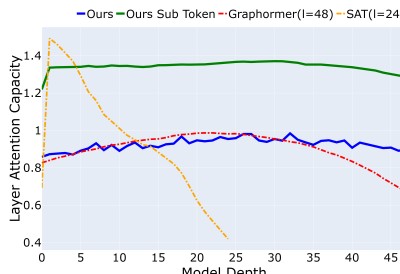

Figure 1: Left: Performance on ZINC dataset of different graph transformers by varying their depths. Our DeepGraph successfully scales up the depth, while the baselines can not. (Lower is better.) Right: Layer attention capacity to substructures with depth.

(Chen et al., 2022), i.e., substructures of a graph, and obtaining particular features. Substructures are the basic intrinsic features of graph data widely used in data analysis (Yu et al., 2020) and machine learning (Shervashidze et al., 2009), as well as graph model interpretability (Miao et al., 2022). Although the self-attention module appears to be very beneficial for automatically learning important substructure features in the graph, our analysis indicates that this ability vanishes as depth grows, restricting the deeper graph transformer from learning useful structure features. Specifically, we focus on the influence of attention on different substructures, which we found to decrease after each self-attention layer. In consequence, it is difficult for deep models to autonomously learn effective attention patterns of substructures and obtain expressive graph substructure features.

We further propose a graph transformer model named DeepGraph with a simple but effective method to enhance substructure encoding ability of deeper graph transformer. The proposed model explicitly introduces local attention mechanism on substructures by employing additional substructure tokens in the model representation and applying local attention to nodes related to those substructures. Our method not only introduces the substructure based attention to encourage the model to focus on substructure feature, but also enlarges the attention capacity theoretically and empirically, which improves the expressiveness of representations learned on substructures.

In summary, our contributions are as follows:

- We present the bottleneck of graph transformers' performance when depth increases, illustrating the depth limitation of current graph transformers. We study the bottleneck from the perspective of attention capacity decay with layers theoretically and empirically, and demonstrate the difficulty for deep models to learn effective attention patterns of substructures and obtain informative graph substructure features.
- According to the above finding, we propose a simple yet effective local attention mechanism based on substructure tokens, promoting focus on local substructure features of deeper graph transformer and improving the expressiveness of learned representations.
- Experiments show that our method unblocks the depth limitation of graph transformer and achieves state-of-the-art results on standard graph benchmarks with deeper models.

## 2 RELATED WORK

**Graph transformers** Transformer with the self-attention has been the mainstream method in nature language processing (Vaswani et al., 2017; Devlin et al., 2019; Liu et al., 2019), and is also proven competitive for image in computer vision (Dosovitskiy et al., 2020). Pure transformers lack relation information between tokens and need position encoding for structure information. Recent works apply transformers in graph tasks by designing a variety of structure encoding techniques. Some works embed structure information into graph nodes by methods including Laplacian vector, random walk, or other feature (Zhang et al., 2020; Dwivedi & Bresson, 2020; Kreuzer et al., 2021; Kim et al., 2022; Wu et al., 2021). Some other works introduce structure information into attention by graph distance, path embedding or feature encoded by GNN (Park et al., 2022; Maziarka et al., 2020; Ying et al., 2021; Chen et al., 2022; Mialon et al., 2021; Choromanski et al., 2022). Other works use transformer as a module of the whole model (Bastos et al., 2022; Guo et al., 2022).

**Deep neural networks** There are many works focus on solving obstacles and release potential of deep neural networks in feed-forward network (FNN) (Telgarsky, 2016; Yarotsky, 2017) and convolutional neural network (CNN) (He et al., 2016a; Simonyan & Zisserman, 2014; Xiao et al., 2018; He et al., 2016b). For graph neural networks, early studies reveal severe performance degradation when stacking many layers (Li et al., 2019; Xu et al., 2018b; Li et al., 2018), caused by problems including over-smoothing and gradient vanishing (Oono & Suzuki, 2019; Zhao & Akoglu, 2019; Rong et al., 2019; Nt & Maehara, 2019). Recent works alleviate these problems by residual connection, dropout, and other methods (Chen et al., 2020a; Li et al., 2020; 2021), and deepen GNN into hundreds and even thousands of layers with better performance. Deep transformer is also proved powerful in nature language processing (Wang et al., 2022) and computational vision (Zhou et al., 2021), while better initialization (Zhang et al., 2019) or architecture (Wang et al., 2019; Liu et al., 2020a; Bachlechner et al., 2021) are proposed to solve optimization instability and over-smoothing (Shi et al., 2021; Wang et al., 2021; Dong et al., 2021).

**Graph substructure** Substructure is one of the basic intrinsic features of graph data, which is widely used in both graph data analysis (Shervashidze et al., 2009; Yanardag & Vishwanathan, 2015; Rossi et al., 2020) and deep graph neural models (Chen et al., 2020b; Bouritsas et al., 2022; Bodnar et al., 2021; Zhang & Li, 2021; Zhao et al., 2021). In application, substructure related methods are widely used in various domain including computational chemistry (Murray & Rees, 2009; Jin et al., 2018; 2019; Duvenaud et al., 2015; Yu et al., 2020), computational biology (Koyutürk et al., 2004) and social network (Jiang et al., 2010). Certain substructures can also be the pivotal feature for graph property prediction, which is a fundamental hypothesis in graph model interpretability studies (Ying et al., 2019; Miao et al., 2022), helping to understand how a graph model makes decisions.

## 3 Preliminary

### 3.1 Graph and Substructure

We denote graph data as $\{G_i, y_i\}$, where a graph $G = (N, R, x, r)$ includes nodes $N = \{1 \ldots |G|\}$ and corresponding edges $R \subset N \times N$, while $x$ and $r$ are node features and edge features. Label $y$ can be graph-wise or node-wise, depending on the task definition. Given graph $G$, a substructure is defined as $G^S = \{N^S, R^S, x^S, r^S\}$, where $N^S \subset N, R^S = (N^S \times N^S) \cap R$, i.e. nodes of $G^S$ form a subset of the graph $G$ and edges are all the existing edges in $G$ between nodes subset, which is also known as induced subgraph. Because attention is only applied to graph nodes, attention to arbitrary subgraphs is not well-defined. Therefore, we only consider induced subgraphs in this work.

### 3.2 Transformer

The core module of the transformer is self-attention. Let $H_l = [h_1, \cdots, h_n]^\top \in \mathbb{R}^{n \times d_h}$ be the hidden representation in layer $l$, where $n$ is the number of token, and $d_h$ is the dimension of hidden embedding of each token. The self-attention mapping $\hat{\text{Attn}}$ with parameter $W^V, W^Q, W^K \in \mathbb{R}^{d_h \times d_k}$ and $W^O \in \mathbb{R}^{d_k \times d_h}$ is

$$\hat{A} = \frac{(H_l W^Q)(H_l W^K)^\top}{\sqrt{d_k}}, \quad A = \text{softmax}(\hat{A}),$$
$$\hat{\text{Attn}}(H_l) = A H_l W^V W^O = A H_l W^{VO}, \tag{1}$$

where $W^{VO} \triangleq W^V W^O$. In practice, a complete transformer layer also contains a two-layer fully-connected network FFN, and layer normalization with residual connection LN is also applied to both self-attention module and fully-connected network:

$$H_{l+1} = \text{FFN}(\text{Attn}(H_l)), \tag{2}$$

where $\text{Attn}(H) = \text{LN}(A H W^{VO})$, $\text{FFN}(H') = \text{LN}(\text{ReLU}(H' W^{F_1} + \mathbf{1} b^{F_1 T}) W^{F_2} + \mathbf{1} b^{F_2 T})$. $\text{LN}(f(X)) = (X + f(X) - \mathbf{1} b^{N T}) D$ is the layer normalization, where $D$ is diagonal matrix with normalizing coefficients.

For graph transformer, structure information can be encoded into token representations or attentions. Our model adopts the distance and path-based relative position encoding methods as Ying et al. (2021), using graph distance DIS and shortest path information SP as relative position:

$$\hat{A}_{ij} = \frac{\left(h_i W^Q\right)\left(h_j W^K\right)^T}{\sqrt{d_k}} + b^D_{\text{DIS}(i,j)} + \underset{k \in \text{SP}(i,j)}{\text{Mean}} b^R_{r_k} \tag{3}$$

We also employ deepnorm (Wang et al., 2022) residual connection method in out model, which adjust the residual connection in layer normalization by constant $\eta$: $\text{LN}(f(X)) = (\eta X + f(X) - \mathbf{1}b^{N^T})D$ to stabilize the optimization of deep transformers.

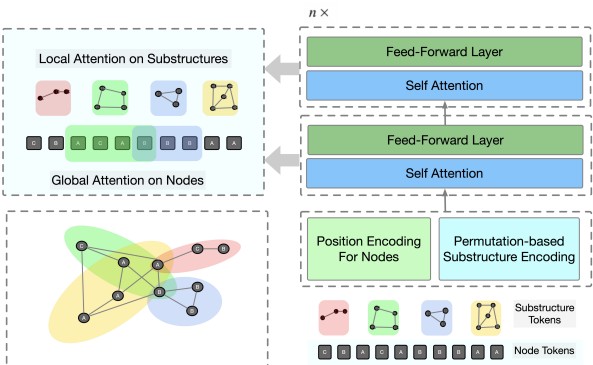

Figure 2: Overview of the proposed graph encoding framework.

## 4 THEORETICAL RESULTS

In this section, we study the capacity of global attention to explain the depth bottleneck in graph transformers. The global attention is intended to focus on important substructures and learn expressive substructure representations automatically. However, the fact that graph transformers perform poorly when more layers are stacked raises doubts about the effectiveness of these self-attention layers. The attention layers require sufficient capacity to represent various substructures, which is necessary for learning attention patterns. We define attention capacity, analyze its variation with depth, and propose that local attention to substructures is a potential solution.

### 4.1 DEFINITION OF ATTENTION CAPACITY

We define the capacity of attention as the maximum difference between representations learned by different substructure attention patterns. Let $\text{supp}(e)$ be all the non-zero dimensions of vector $e$. We define a fixed attention pattern $e$ of substructure $G^S$ as an attention vector that only focuses on nodes of $G^S$, i.e., $\text{supp}(e) = N^S$, where $e \in [0,1]^n, e^T \mathbf{1} = 1$.

Given a graph $G$ with several important substructures $G^S_1 \ldots G^S_m$ and corresponding attention patterns $e_1 \ldots e_m$, denote $E = (e_1, e_2 \ldots, e_m)$ where columns of $E$ are the base vectors of attention patterns on substructures. We consider the attention space spanned by these attention patterns. Attention from this space only focuses on the important substructures:

$$\Delta_S \triangleq \{Ec | c \in [0,1]^m, c^T \mathbf{1} = 1\}, \tag{4}$$

Denote $\Delta^n_S$ as matrix space with n columns all in space $\Delta_S$. We then define attention capacity as the maximum difference between outputs computed by the self-attention with different attention matrices from space $\Delta^n_S$. Let $\hat{\text{Attn}}_A$ be the self-attention mapping where attention matrix equals $A$.

**Definition 1** *The attention capacity is defined as:*

$$\begin{aligned}
F_H &= \max_{A_1^T, A_2^T \in \Delta^n_S} |\hat{\text{Attn}}_{A_1}(H) - \hat{\text{Attn}}_{A_2}(H)|_F \\
&= \max_{C_1, C_2 \in \{C | C \in [0,1]^{m \times n}, C^T \mathbf{1} = \mathbf{I}\}} |C_1^T E^T H W^{VO} - C_2^T E^T H W^{VO}|_F,
\end{aligned} \tag{4.1}$$

Attention capacity in graph transformers is crucial. Larger capacity enables varying features by focusing on different substructures, while smaller capacity limits attention patterns' impact on learned representations. Smaller capacity modules have difficulty learning attention patterns as they are less sensitive to substructures.

## 4.2 ATTENTION CAPACITY DECREASES WITH DEPTH

After defining the measure of attention capacity for substructures, we investigate the attention capacity of graph transformers w.r.t depth both empirically and theoretically. We first empirically demonstrate how the attention capacity in the graph transformer varies with depth, then theoretically analyze the phenomenon and possible solution.

The attention capacity of deep graph transformers for various substructures is computed per Definition 4.1 and normalized by hidden embedding norm (see Appendix G). Figure 1 (right) shows the results. Graphormer (Ying et al., 2021)'s attention capacity decreases significantly after the 24th layer, while SAT (Chen et al., 2022) decreases across all layers, indicating their limitations.

Next, we illustrate that the stacked self-attention layers can cause decreased attention capacity for deeper layers. We first demonstrate that the self-attention module decreases the capacity of attention, then discuss the case with fully-connected layers.

**Theorem 1 (The stacked attention modules decrease the capacity of attention)** *We analyze capacity of attention $F_{H_l}$ for stacked attention modules $H_{i+1} = Attn(H_i) = (H_i + A_i H_i W_i^{VO} - \mathbf{1}b_i^{N^T})D_i$. Assume $H_i W_i^{VO} D_i$ is full row rank for each layer $i$. Due to the property of layer normalization and properly designed initialization, assume the output of attention module equals to its input, i.e., for input $X$, $|\text{Attn}(X)|_F \approx |X|_F$. Then the attention capacity of layer $l$ is upper bounded by*

$$F_{H_l} \leq \sqrt{2m} \prod_{i=1}^{l-1} (\alpha_i) |P_m E^T|_2 |W_l^{VO}|_2 |PH_1|_F \tag{5}$$

*,where $\alpha_i = \frac{|(PH_i + PA_i H_i W_i^{VO})D_i|_F}{|(PH_i + A_i PH_i W_i^{VO} - \mathbf{1}b_i^{N^T})D_i|_F} < 1$ with probability 1, and $P \triangleq (I - \frac{1}{n}\mathbf{1}\mathbf{1}^T)$, $P_m \triangleq (I - \frac{1}{m}\mathbf{1}\mathbf{1}^T)$.*

Proof is in Appendix 1. The analysis above reveals that the self-attention module decreases the upper bound of attention capacity exponentially. We next consider the case when fully-connected layers are also included in each layer, just like the architecture in practice.

**Theorem 2 (The upper bound of attention capacity after stacked transformer layers)** *For transformer layer with self-attention $H_i' = \text{Attn}(H_i) = (H_i + A_i H_i W_i^{VO} - \mathbf{1}b_i^{N^T})D_i$, and fully-connected layer $H_{i+1} = \text{FFN}(H_i') = (H_i' + ReLU(H_i' W_i^{F_1} + \mathbf{1}b_i^{F_1^T})W_i^{F_2} + \mathbf{1}b_i^{F_2^T} - \mathbf{1}b_i^{N_2^T})D_i'$, with the same assumption as the previous, attention capacity of layer $l$ is upper bounded as follows:*

$$F_{H_l} \leq \sqrt{2m} \prod_{i=1}^{l-1} (\alpha_i \gamma_i) |P_m E^T|_2 |W_l^{VO}|_2 |PH_1|_F \tag{6}$$

*,where $\alpha_i = \frac{|(PH_i + PA_i H_i W_i^{VO})D_i|_F}{|(PH_i + A_i PH_i W_i^{VO} - \mathbf{1}b_i^{N^T})D_i|_F} < 1$ with probability 1, and $\gamma_i = |D_i'|_2(1 + |W_i^{F_1}|_2|W_i^{F_2}|_2)$.*

Proof in Appendix 2. Fully-connected layer impact can be bounded by coefficients $\gamma_i$, which describe hidden embedding norm change. Previous work Shi et al. (2021) shows that $\gamma_i$ is dominated by $|D_i'|_2$, and for a fraction of data $|D_i'|_2 < 1$, leading to attention capacity vanishing in these cases.

## 4.3 LOCAL ATTENTION FOR DEEP MODEL

Reducing attention capacity with depth creates two problems for graph transformers: difficulty attending to substructures and loss of feature expressiveness to substructures. To address the first problem, it's natural to introduce substructure-based local attention as inductive bias into graph transformer to make up for the deficiency of attention on substructures, where each node can only attend to other nodes that belong to same substructures. Furthermore, We will next show that introducing substructures based local attention also addresses the second problem. We next prove that capacity decay can be alleviated if local attention to substructures is applied in each layer.

**Theorem 3** *Define substructure based local attention as where each node only attends nodes that belong to the same substructures. Assume that for each node, the number of nodes belonging to the same substructures is at most $r$, where $r < n$. Let $\alpha^s$ be the decay coefficient of this model in Theorem 1, and $\alpha$ be the decay coefficient of the global attention model. Denoted minimum of $\alpha^s$ and $\alpha$ for all possible representation $H_i$ and model parameters by $\alpha^{s*}$ and $\alpha^*$ respectively. Then we have $\alpha^{s*} > \alpha^*$.*

The proof is in Appendix 3. This theory illustrates that substructure based local attention not only introduces the substructure based attention to encourage the model to focus on substructure features but also enlarges the attention capacity of deep layers, which promotes the representation capacity of the model when it grows deep.

## 5 APPROACH

Replacing global attention directly with local attention may lead to insufficient learning ability of the model for long-range dependencies. To better introduce local attention to substructure in self-attention models, we propose a novel substructure token based local attention method, DeepGraph, as Figure 2 shows. DeepGraph tokenizes the graph into node level and substructure level. Local attention is applied to the substructure token and corresponding nodes, while the global attention between nodes is still preserved. This provides a better solution for explicitly introducing attention to substructures in the model while combining global attention, to achieve a better global-local encoding balance.

Our method first samples substructures of the graph, then encodes substructures into tokens, and finally enforces local attention on substructures.

### 5.1 SUBSTRUCTURE SAMPLING

As we aim to stress critical substructures in graph, there are numerous types of substructures that are categorized into two groups: neighbors containing local features, such as k-hop neighbors Zhang & Li (2021) and random walk neighbors Zhao et al. (2021), and geometric substructures representing graph topological features, such as circles Bodnar et al. (2021), paths, stars, and other special subgraphs Bouritsas et al. (2022). We integrate these substructures into a unified substructure vocabulary in this work. In order to match geometric substructures, we use the Python package graph-tool that performs efficient subgraph matching. It is practical to pre-compute these substructures once, and then cache them for reuse. It takes about 3 minutes to compute all the substructures of ZINC, and about 2 hours for PCQM4M-LSC.

Substructures in a graph often exceed the number of nodes. In order to ensure the feasibility of the computation, we sample substructures in each computation. Formally, at each time of encoding, for a graph $G$ with the corresponding substructure set $\{G^S\}$, we sample a subset $\{G_1^S, G_2^S \ldots G_m^S\}$ as input to our model:

$$\{G_1^S, G_2^S \ldots G_m^S\} = \text{SubstructureSampling}(\{G^S\}),$$
$$\hat{y} = \text{DeepGraph}(G, \{G_1^S, G_2^S \ldots G_m^S\})$$

(7)

We hope the sampled substructures cover every node of the graph as evenly as possible (Zhao et al., 2021) in order to reduce biases resulting from the uneven density of substructures. We also balance the types of substructures in our sampling, due to the divergent ratios of different substructure types in a graph. The details are in Appendix C and D.

### 5.2 SUBSTRUCTURE TOKEN ENCODING

The input embedding contains node tokens embedding $\{h_1, h_2, \ldots, h_n\}$ and substructure tokens embedding $\{h_{n+1}, \ldots, h_{n+m}\}$, encoded by node feature encoder $g_n$ and substructure encoder $g_s$ individually. The token embedding of graph nodes is mapped from graph node feature $x$ to feature vector $h$: $h_i = g_n(x_i), i \in \{1, 2, \ldots n\}$.

For the substructure tokens, we apply permutation-based structure encoding (Murphy et al., 2019; Chen et al., 2020b; Nikolentzos & Vazirgiannis, 2020) by encoding the substructure adjacency matrix $\mathbf{A_i^S}$ directly. The core idea of permutation-based encoding is pooling the encoding output of

permuted graph information over all the possible permutations, which is permutation-invariant and has no information loss. We further apply depth-first search (DFS) with the order of node degree to reduce the possible order of substructure nodes. At the beginning and at each step of DFS, we use degrees to sort nodes. A random node from the ones with the least degree is selected as the starting point for DFS. The nodes are sorted according to their degree at each DFS step, while the same-degree nodes are randomly permuted. See Appendix E for more details. Using this method, the graph encoding permutations is invariant and the number of possible combinations is greatly reduced. The formal definition of substructure token encoder is

$$h_{n+i} = \underset{\pi \in \text{DFS}(\mathbf{A_i^S})}{Pool} g_s(\pi(\mathbf{A_i^S})), i \in \{1, 2, \ldots m\} \tag{8}$$

In practice, sampling is applied instead of pooling. A single sample is sufficient during training to allow the model to learn the substructure stably.

## 5.3 LOCAL ATTENTION ON SUBSTRUCTURES

The substructure and its corresponding nodes receive localized attention after substructure tokens have been added. Given input embedding $H_1 = (h_1, h_2, \ldots, h_{n+m})$, mask $M$ is added in self-attention module $\hat{\text{Attn}}_m(H, M)$ as

$$\hat{\text{Attn}}_m(H_l, M) = \text{softmax}(\hat{A} + M)H_l W^{VO} \tag{9}$$

where $M_{ij} \in \{0, -\infty\}$, i.e., the elements of mask $M$ can be 0 or $-\infty$, leading to a sparse attention matrix $A$ with zero in position of $-\infty$ in $M$. In our model, the mask is defined as follows to induce local attention to substructure tokens and corresponding nodes: $M_{ij} = -\infty$ if $i + n \in \{n+1, n+2, \ldots n+m\}, j \notin N_i^S$ or $j + n \in \{n+1, n+2, \ldots n+m\}, i \notin N_j^S$, and $M_{ij} = 0$ otherwise. The substructure tokens only apply local attention to corresponding nodes.

The attention in our model is a combination of local attention on substructures and global attention. Substructure tokens are the core of local attention, attending only to corresponding nodes, and integrating representation from the whole substructure. Nodes belonging to the same substructure share a substructure token message, increasing distinguishability of substructures and promoting substructure representation capacity.

## 6 EXPERIMENTS

In this section, we aim to validate the performance of DeepGraph empirically. Specifically, we attempt to answer the following questions: **(i)** How does DeepGraph perform in comparison to existing transformers on popular benchmarks? **(ii)** Does DeepGraph's performance improve with increasing depth? **(iii)** Is DeepGraph capable of alleviating the problem of shrinking attention capacity? **(IV)** What is the impact of each part of the model on overall performance? We first conduct experiments to evaluate our model on four popular graph datasets, comparing it with state-of-the-art graph transformer models, as well as their augmented deeper versions. Then we illustrate the attention capacity of DeepGraph by visualization. Finally, we validate the effect of different components through ablation studies. Codes are available at https://github.com/zhao-ht/DeepGraph.

## 6.1 DATASETS

Our method is validated on the tasks of the graph property prediction and node classification, specifically including PCQM4M-LSC (Hu et al., 2020), ZINC (Dwivedi et al., 2020), CLUSTER (Dwivedi et al., 2020) and PATTERN (Dwivedi et al., 2020), widely used in graph transformer studies. PCQM4M-LSC is a large-scale graph-level regression dataset with more than 3.8M molecules. ZINC consists of 12,000 graphs with a molecule property for regression. CLUSTER and PATTERN are challenging node classification tasks with graph sizes varying from dozens to hundreds, containing 14,000 and 12,000 graphs, respectively.

## 6.2 BASELINES

We choose recent state-of-art graph transformer models: GT (Dwivedi & Bresson, 2020), SAN (Kreuzer et al., 2021), Graphormer (Ying et al., 2021) and SAT (Chen et al., 2022), covering various

| | Graph regression | | Node classification | |
|---|---|---|---|---|
| | PCQM4M-LSC ↓ | ZINC ↓ | CLUSTER ↑ | PATTERN ↑ |
| GCN | 0.1691 | 0.367 ± 0.011 | 68.498 ± 0.976 | 71.892 ± 0.334 |
| GIN | 0.1537 | 0.526 ± 0.051 | 64.716 ± 1.553 | 85.387 ± 0.136 |
| GT-sparse | - | 0.226 ± 0.014 | 73.169 ± 0.622 | 84.808 ± 0.068 |
| GT-Full | - | 0.598 ± 0.049 | 27.121 ± 8.471 | 56.482 ± 3.549 |
| SAN-Sparse | - | 0.198 ± 0.004 | 75.738 ± 0.106 | 81.329 ± 2.150 |
| SAN-Full | - | 0.139 ± 0.006 | 76.691 ± 0.247 | 86.581 ± 0.037 |
| Graphormer | 0.1234 | 0.122 ± 0.006 | - | - |
| SAT | - | 0.094 ± 0.008 | 77.856 ± 0.104 | 86.865 ± 0.043 |
| DeepGraph (12) | 0.1220 | 0.078 ± 0.006 | 77.526 ± 0.122 | 90.015 ± 0.038 |
| DeepGraph (24) | 0.1206 | 0.076 ± 0.004 | 77.810 ± 0.167 | 90.522 ± 0.059 |
| DeepGraph (48) | **0.1193** | **0.072 ± 0.004** | **77.912 ± 0.138** | **90.657 ± 0.062** |

Table 1: Comparison of our DeepGraph to SOTA methods on graph regression and node classification tasks.

graph transformer methods including absolute and relative position embedding. Graph neural networks baselines includes GCN (Kipf & Welling, 2016) and GIN (Xu et al., 2018a). We first compare our model with the standard state-of-art models. Then we compare our model with the deepened state-of-arts augmented by recent algorithms for deep transformers, including the fusion method (Shi et al., 2021), and reattention method (Zhou et al., 2021). As we use deepnorm to stabilize our training process, we compare DeepGraph to naive deepnorm in the ablation study.

## 6.3 SETTINGS

We implement DeepGraph with 12, 24, and 48 layers. The hidden dimension is 80 for ZINC and PATTERN, 48 for CLUSTER, and 768 for PCQM4M-LSC. The training uses Adam optimizer, with warm-up and decaying learning rates. Reported results are the average of over 4 seeds. Both geometric substructure and neighbors are used as substructure patterns in our model. We imply both geometric substructures and k-hop neighbors on ZINC and PCQM4M-LSC. As for CLUSTER and PATTERN, only random walk neighbors are used due to the large scale and dense connection of these two datasets. The effect of different substructure patterns is shown in the ablation study.

## 6.4 MAIN RESULTS

Table 1 summarizes our experimental results on the graph regression and node classification tasks. The data for the baseline models are taken from their papers. The depth of our model varies from 12 to 48 layers. As the results illustrated, our 12-layer model outperforms baseline models on PCQM4M-LSC, ZINC, and PATTERN, especially on ZINC and PATTERN, surpassing the previous best result reported significantly. As model depth increases, our model achieves consistent improvement. Note that our model with 48 layers outperforms baseline models on all the datasets, proving the effectiveness of more stacked DeepGraph layers. The parameter number of each model for each dataset is provided in Appendix H.

## 6.5 EFFECT OF DEEPENING

We next compare DeepGraph with other deepened baseline models. We choose Graphormer and SAT as baseline models, and we deepen them by 2 and 4 times compared to the original version. Along with deepening naively, we also enhance baseline models through the fusion method and reattention method.

| | DeepGraph 12 | | DeepGraph 48 | |
|---|---|---|---|---|
| | ZINC | CLUSTER | ZINC | CLUSTER |
| - local attention | 0.135 | 76.167 | 0.141 | 76.133 |
| - substructure encoding | 0.085 | 76.531 | 0.124 | 77.884 |
| - deepnorm | 0.080 | 77.202 | 0.074 | 77.682 |
| DeepGraph | **0.078** | **77.526** | **0.072** | **77.912** |

| | DeepGraph 12 | DeepGraph 48 |
|---|---|---|
| Geometric only | 0.078 | 0.075 |
| Neighbors only | 0.086 | 0.079 |
| Both | **0.078** | **0.072** |

Table 2: Ablation study of local attention, substructure encoding, and deepnorm on ZINC and CLUSTER.

Table 3: Sensitivity study of different substructure types on ZINC.

Figure 3: Comparison of DeepGraph to SOTA methods deepened by different deep transformer methods.

As shown in Figure 3, DeepGraph outperforms baselines by a significant margin. Naive deepening decreases performance on all the datasets for both baseline models. Note that all the 4-time deep SAT fail to converge on CLUSTER due to the training difficulty, as well as Graphormer with reattention. Fusion improves deeper Graphormer on PCQM4M-LSC, and reattention boosts 2x deep SAT on CLUSTER slightly, but none of them are consistent. Our methods surpass fusion and reattention methods stably on all 4 datasets, supporting our theoretical claim that sensitivity to substructures is crucial in deep graph transformers.

## 6.6 VISUALIZATION OF ATTENTION CAPACITY

We visualize attention capacity of our method and deep baselines on ZINC dataset, using the same substructures as our model. Additionally, to illustrate the capacity of token representation of substructures, we also directly compute the maximum difference between all the substructure tokens value vectors. All the results are normalized by the corresponding representation norm. See Appendix G for details. The results are plotted in Figure 1 (right).

First, the attention capacity of our model remains high across all layers as compared to baselines, indicating that substructure-based local attention can be a useful method for preventing attention deficits. Second, substructure tokens have a much higher capacity than attention capacity computed on nodes, demonstrating the benefits of using substructure tokens for substructure encoding.

## 6.7 ABLATION AND SENSITIVITY STUDY

Finally, we verify our model's effectiveness through an ablation study on ZINC and CLUSTER tasks, removing local attention, substructure encoding, and deepnorm to observe their impact. Note that without local attention, the model is only a relative position-based graph transformer with deepnorm residual connections. Without substructure encoding, randomly initialized embedding is applied to each substructure token. Results in Table 2 show that without local attention, performance decreases significantly on both datasets, especially for deeper models, proving the effectiveness of the proposed methods. Furthermore, substructure encoding also plays an instrumental role, emphasizing the importance of structural information. Finally, deepnorm also contributes to model performance by stabilizing the optimization, especially for 48-layer models.

We validate the sensitivity of our model to different substructure types by testing the performance of DeepGraph with only geometric substructures or neighbors. Table 3 indicates that both contribute to performance, but geometric substructures are more critical for ZINC. This result is expected because specific structures like rings are important fundamental features for the molecule data, and it can be further strengthened when combined with neighbor substructures. Our model can flexibly use different forms of substructure to fully utilize prior knowledge of data.

## 7 CONCLUSIONS

This work presents the bottleneck of graph transformers' performance when depth increases. By empirical and theoretical analysis, we find that deep graph transformers are limited by the capacity bottleneck of graph attention. Furthermore, we propose a novel graph transformer model based on substructure-based local attention with additional substructure tokens. Our model enhances the ability of the attention mechanism to focus on substructures and addresses the limitation of encoding expressive features as the model deepens. Experiments show that our model breaks the depth bottleneck and achieves state-of-the-art performance on popular graph benchmarks.

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

# A  PROOF OF THEOREMS

**Proof 1** *for brevity, define $P \triangleq (I - \frac{1}{n}\mathbf{1}\mathbf{1}^T)$, and $P_m \triangleq (I - \frac{1}{m}\mathbf{1}\mathbf{1}^T)$. We first introduce an important property of $P$: For any matrix $B$ with left eigenvector $B\mathbf{1} = \beta\mathbf{1}$, we have $PBC=PBPC, \forall C$. To prove it, we can deompose $C = PC + \frac{1}{n}\mathbf{1}\mathbf{1}^T C$. Then*

$$PBC = PB(PC + \frac{1}{n}\mathbf{1}\mathbf{1}^T C) = PBPC + \beta\frac{1}{n}P\mathbf{1}\mathbf{1}^T C = PBPC$$

*We next prove the main theorem by recursion. By definition,*

$$
\begin{aligned}
F_H &= \max_{A_1^T, A_2^T \in \Delta_S^n} |\hat{Attn}_{A_1}(H) - \hat{Attn}_{A_2}(H)|_F \\
&= \max_{C_1, C_2 \in \{C | C \in [0,1]^{m \times n}, C^T\mathbf{1}=\mathbf{1}\}} |C_1^T E^T HW^{VO} - C_2^T E^T HW^{VO}|_F \\
&= \max_{C_1, C_2 \in \{C | C \in [0,1]^{m \times n}, C^T\mathbf{1}=\mathbf{1}\}} |(C_1 - C_2)^T E^T HW^{VO}|_F
\end{aligned}
\tag{10}
$$

*Note that $(C_1 - C_2)^T\mathbf{1} = \mathbf{1} - \mathbf{1} = 0$, so we can multiply $P_m$ before it:*

$$
\begin{aligned}
F_H &= \max_{C_1, C_2 \in \{C | C \in [0,1]^{m \times n}, C^T\mathbf{1}=\mathbf{1}\}} |(C_1 - C_2)^T E^T HW^{VO}|_F \\
&= \max_{C_1, C_2 \in \{C | C \in [0,1]^{m \times n}, C^T\mathbf{1}=\mathbf{1}\}} |(C_1 - C_2)^T P_m E^T HW^{VO}|_F \\
&\le \max_{C_1, C_2 \in \{C | C \in [0,1]^{m \times n}, C^T\mathbf{1}=\mathbf{1}\}} |(C_1 - C_2)^T|_2 |P_m E^T HW^{VO}|_F \\
&\le \sqrt{2m} |P_m E^T HW^{VO}|_F
\end{aligned}
\tag{11}
$$

*,where $|(C_1 - C_2)^T|_2 \le \sqrt{|(C_1 - C_2)^T|_1 |(C_1 - C_2)^T|_\infty} = \sqrt{2m}$.*

*We further decompose $|P_m E^T HW^{VO}|_F$ into*

$$|P_m E^T HW^{VO}|_F = |P_m E^T PHW^{VO}|_F \le |P_m E^T|_2 |W^{VO}|_2 |PH|_F \tag{12}$$

*For layer $l$, we recursively compute $|PH_{i+1}|_F$ by $|PH_i|_F$ for layer $i = l-1, l-2, \ldots, 1$:*

$$
\begin{aligned}
|PH_{i+1}|_F &= |P(H_i + A_i H_i W_i^{VO} - \mathbf{1}b_i^{N^T})D_i|_F \\
&= |P(PH_i + A_i PH_i W_i^{VO} - \mathbf{1}b_i^{N^T})D_i|_F \\
&= \frac{|P(PH_i + A_i PH_i W_i^{VO} - \mathbf{1}b_i^{N^T})D_i|_F}{|(PH_i + A_i PH_i W_i^{VO} - \mathbf{1}b_i^{N^T})D_i|_F} \frac{|(PH_i + A_i PH_i W_i^{VO} - \mathbf{1}b_i^{N^T})D_i|_F}{|PH_i|_F} |PH_i|_F \\
&= \alpha_i \lambda_i |PH_i|_F
\end{aligned}
\tag{13}
$$

*,where $\alpha_i = \frac{|P(PH_i + A_i PH_i W_i^{VO} - \mathbf{1}b_i^{N^T})D_i|_F}{|(PH_i + A_i PH_i W_i^{VO} - \mathbf{1}b_i^{N^T})D_i|_F}$, and $\lambda_i = \frac{|(PH_i + A_i PH_i W_i^{VO} - \mathbf{1}b_i^{N^T})D_i|_F}{|PH_i|_F}$.*

*Note that $P = (I - \frac{1}{n}\mathbf{1}\mathbf{1}^T)$ is a contraction mapping, so $\alpha_i \le 1$, and equal to 1 if and only if $A_i PH_i W_i^{VO} D_i - \mathbf{1}b_i^{N^T} D_i$ is orthogonal to the $\mathbf{1}$-stretched space, which is usually impossible, because $-\mathbf{1}b_i^{N^T} D_i$ is in the $\mathbf{1}$-stretched space, and the probability that projection of $A_i PH_i W_i^{VO} D_i$ on $\mathbf{1}$-stretched space equals to $\mathbf{1}b_i^{N^T} D_i$ is 0 due to the property of random matrix. We further show that $A_i PH_i W^{VO} D_i$ is full-rank with probability 1: As assumption, $H_i W_i^{VO} D_i$ is full-rank, and $\mathbf{1}$-stretched space is the only subspace that $PH_i W_i^{VO} D_i$ is orthogonal to, i.e.*

$$x^T PH_i W_i^{VO} D_i = 0 \Leftrightarrow x = \beta\mathbf{1}$$

. So if $\mathbf{1}^T A_i P H_i W_i^{VO} D_i = 0$, $\mathbf{1}^T A_i = \beta \mathbf{1}^T$, i.e. $A_i^T \mathbf{1} = \beta \mathbf{1}$, the probability of which is 0 for random parameters in transformer.

For coefficient $\lambda_i$, due to the property of layer normalization and good parameter initialization that keeps norm of output equals to input, i.e. $|(X + A_i X W_i^{VO} - \mathbf{1} b_i^N) D_i|_F / |X|_F \approx 1, \forall X$, we have $\lambda \approx 1$.

Based on the analysis above, we can bound $F_{H_l}$ recursively:

$$F_{H_l} \leq \sqrt{2m} \prod_{i=1}^{l-1} (\alpha_i) |P_m E^T|_2 |W_l^{VO}|_2 |PH_1|_F \tag{14}$$

, where $\alpha_i = \frac{|P(PH_i + A_i PH_i W_i^{VO} - \mathbf{1} b_i^{N^T}) D_i|_F}{|(PH_i + A_i PH_i W_i^{VO} - \mathbf{1} b_i^{N^T}) D_i|_F} = \frac{(|PH_i + P A_i H_i W_i^{VO}) D_i|_F}{|(PH_i + A_i PH_i W_i^{VO} - \mathbf{1} b_i^{N^T}) D_i|_F} < 1$ with probability 1.

**Proof 2** As the same method in the previous proof, we need to recursively bound $|PH_{i+1}|_F$ by $|PH_i|_F$ for layer $i = l-1, l-2, \ldots, 1$. Deote $H_i' = \text{Attn}(H_i)$, and $H_{i+1} = \text{FFN}(H_i')$. Similar to the previous proof, we have

$$
\begin{aligned}
|PH_i'|_F &= |P(H_i + A_i H_i W_i^{VO} - \mathbf{1} b_i^{N^T}) D_i|_F \\
&= |P(PH_i + A_i PH_i W_i^{VO} - \mathbf{1} b_i^{N^T}) D_i|_F \\
&= \frac{|P(PH_i + A_i PH_i W_i^{VO} - \mathbf{1} b_i^{N^T}) D_i|_F}{|(PH_i + A_i PH_i W_i^{VO} - \mathbf{1} b_i^{N^T}) D_i|_F} \frac{|(PH_i + A_i PH_i W_i^{VO} - \mathbf{1} b_i^{N^T}) D_i|_F}{|PH_i|_F} |PH_i|_F \\
&= \alpha_i \lambda_i |PH_i|_F
\end{aligned}
\tag{15}
$$

As for $PH_{i+1}$, we can bound it as

$$
\begin{aligned}
|PH_{i+1}|_F &= |P(H_i' + ReLU(H_i' W_i^{F_1} + \mathbf{1} b_i^{F_1^T}) W_i^{F_2} + \mathbf{1} b_i^{F_2^T} - \mathbf{1} b_i^{N_2^T}) D_i'|_F \\
&= |PH_i' D_i'|_F + |P ReLU(H_i' W_i^{F_1} + \mathbf{1} b_i^{F_1^T}) W_i^{F_2} D_i'|_F \\
&\leq |PH_i'|_F |D_i'|_2 + |P ReLU(H_i' W_i^{F_1} + \mathbf{1} b_i^{F_1^T})|_F |W_i^{F_2}|_2 |D_i'|_2 \\
&\leq |PH_i'|_F |D_i'|_2 + |P(H_i' W_i^{F_1} + \mathbf{1} b_i^{F_1^T})|_F |W_i^{F_2}|_2 |D_i'|_2 \\
&\leq |PH_i'|_F |D_i'|_2 + |PH_i'|_F |W_i^{F_1}|_2 |W_i^{F_2}|_2 |D_i'|_2 \\
&= |D_i'|_2 (1 + |W_i^{F_1}|_2 |W_i^{F_2}|_2) |PH_i'|_F
\end{aligned}
\tag{16}
$$

So the final conclusion is

$$F_{H_l} \leq \sqrt{2m} \prod_{i=1}^{l-1} (\alpha_i \gamma_i) |P_m E^T|_2 |W_l^{VO}|_2 |PH_1|_F \tag{17}$$

, where $\alpha_i = \frac{|P(PH_i + A_i PH_i W_i^{VO} - \mathbf{1} b_i^{N^T}) D_i|_F}{|(PH_i + A_i PH_i W_i^{VO} - \mathbf{1} b_i^{N^T}) D_i|_F} = \frac{(|PH_i + P A_i H_i W_i^{VO}) D_i|_F}{|(PH_i + A_i PH_i W_i^{VO} - \mathbf{1} b_i^{N^T}) D_i|_F} < 1$ with probability 1, and $\gamma_i = |D_i'|_2 (1 + |W_i^{F_1}|_2 |W_i^{F_2}|_2)$.

**Proof 3** We denote global attention and local attention by $A$ and $A^s$, respectively.

By definition, $\alpha_i = \frac{(|PH_i + P A_i H_i W_i^{VO}) D_i|_F}{|(PH_i + A_i PH_i W_i^{VO} - \mathbf{1} b_i^{N^T}) D_i|_F}$. We analyse the worst case for $\alpha_i$:

$$
\begin{aligned}
\alpha_i &= \frac{(|PH_i + PA_iH_iW_i^{VO})D_i|_F}{|(PH_i + A_iPH_iW_i^{VO} - \boldsymbol{1}b_i^{N^T})D_i|_F} \\
&= \frac{(|PH_i + PA_iH_iW_i^{VO})D_i|_F}{|(PH_i + (P + \frac{1}{n}\boldsymbol{1}\boldsymbol{1}^T)A_iPH_iW_i^{VO} - \boldsymbol{1}b_i^{N^T})D_i|_F} \\
&\geq \frac{(|PH_i + PA_iH_iW_i^{VO})D_i|_F}{|(PH_i + PA_iH_iW_i^{VO})D_i|_F + |\frac{1}{n}\boldsymbol{1}\boldsymbol{1}^TA_iPH_iW_i^{VO}D_i|_F + |\boldsymbol{1}b_i^{N^T}D_i|_F}
\end{aligned}
\tag{18}
$$

*Further analysis on $|\frac{1}{n}\boldsymbol{1}\boldsymbol{1}^TA_iPH_iW_i^{VO}D_i|_F$ shows that*

$$
\begin{aligned}
|\frac{1}{n}\boldsymbol{1}\boldsymbol{1}^TA_iPH_iW_i^{VO}D_i|_F &= |\boldsymbol{1}^TA_iPH_iW_i^{VO}D_i|_2 \\
&\leq |H_iW_i^{VO}D_i|_2|PA_i^T\boldsymbol{1}|_2,
\end{aligned}
\tag{19}
$$

*where $PA_i^T\boldsymbol{1} = A_i^T\boldsymbol{1} - \boldsymbol{1}$, the element of which is between $[-1, n-1]$, and the sum of all elements of which is 0. The maximum of $|PA_i^T\boldsymbol{1}|_2$ is achieved when one element is $n-1$ and other elements are $-1$.*

*However, for local attention $A_i^s$ that each node is only attended by at most $r$ nodes, where $r < n$, the elements of $|PA_i^{s^T}\boldsymbol{1}|_2$ is only between $[-1, r-1]$, and thus the maximum $|PA_i^{s^T}\boldsymbol{1}|_2$ is less than $|PA_i^T\boldsymbol{1}|_2$.*

*We further address that the two inequalities in 18 and 19 both can be achieved for proper $H_i$ and parameters. So we can conclude that for the worst case of $\alpha_i$ and $\alpha_i^s$, denoted by $\alpha_i^*$ and $\alpha_i^{s*}$,*

$$
\alpha_i^* < \alpha_i^{s*}
$$

## B   DISCUSSION OF SUBSTRUCTURE BASED ATTENTION SPACE

In Definition 4.1, for a substructure $G^S$, the fixed attention pattern $e$ is an ideal attention pattern to help to concentrate on the information of this substructure, for example, a uniform attention vector on a benzene ring, or attention on a two-hop neighbor graph attenuating according to the distance to the central node. Traditional GNN is also an example of such a fixed attention pattern, where each node takes uniform attention to its' neighbor nodes. Although we treat this pattern as fixed on the substructure, we denote that the definition is general because the value can be any learnable attention pattern.

The defined attention space construct attention patterns for all meaningful attention on substructures by containing the combination of all the elementary substructure pattern. As this substructure attention patterns are considered important indicative bias, the graph transformer is expected to learn attention patterns in this space to utilize the substructure feature. This definition is also general and expressive, as sufficient patterns can be added to include more general cases.

## C   SUBSTRUCTURES

The sizes of the different substructures used in this study are neighbor substructures including 2-hop neighbor and 10-step random walk neighbor, and geometric substructures including circle with size from 3 to 8, star with size from 2 to 6, and path with size from 4 to 8. For geometric substructures, we use the python package graph-tool for substructure matching. It takes about 3 minutes to match all the geometric substructures of ZINC, and about 2 hours for PCQM4M-LSC. We cache all the substructures for reuse.

## D   SUBSTRUCTURESAMPLING

A greedy sampling algorithm is used for the substructure sampling process. In each step of the iteration, nodes with less coverage have a greater probability of being covered in the next iteration. At

the beginning of sampling, we randomly select $n_{init}$ substructures from all the substructures, where different types of substructures are balanced. In each iteration, we first mark nodes as set $N_{left}$ that are covered less than threshold $thre$ times by already sampled substructures. Then we calculate $cnt_i$ for each substructure left, which is the number of nodes in $N_{left}$ covered by substructure $G_i^S$. We select $n_{sample}$ from substructures with top $k$ $cnt$ randomly. The iteration ends when each node is covered at least $thre$ times, or no substructure is left. In the algorithm, $n_{init}$, $k$, and $n_{sample}$ are determined by $thre$ and nodes number to speed up the iteration and increase sampling variance, while retaining that nodes are covered as evenly as possible.

## E   DEPTH-FIRST SEARCH (DFS)

We use degree to sort nodes at the beginning of DFS and each step. In the beginning, DFS start from node uniformly sampled from nodes with the least degree. At each step, nodes are sorted according to degree, while nodes with the same degree are permuted randomly. This method keeps the graph encoding permutation invariant, and reduces possible permutations a lot. The drawback of permutation-based encoding is that the number of possible permutations becomes intractable when the graph is large. Fortunately, the substructure size is usually not larger than 10.

## F   RELATED WORK OF TOKEN UNIT

The most suitable basic unit for transformer as a token has been studied in many works in natural language processing and computational vision. For nature language, sentences are separated into sub-words containing a variant number of primary characters (Kudo & Richardson, 2018). In computational vision, while previous works apply transformer directly on image pixels (Parmar et al., 2018; Hu et al., 2019), recent works find it more beneficial to treat patches as tokens (Dosovitskiy et al., 2020), indicating the importance of proper segmentation of the input. For graph transformers, tokens in most current studies are nodes. We propose to encode substructures together with nodes in the graph as tokens in the transformer.

## G   EMPIRICAL STUDY ON ATTENTION CAPACITY

We empirically explore the capacity of the model for the substructure, and validate our theoretical results. We aim to answer the following three questions: (1) Do attention capacity really decreases in deeper graph transformer? (2) Can our method, i.e., local attention with substructure tokens, help to increase model capacity of representing substructures? (3) Can our method, i.e., local attention with substructure tokens, help to alleviate attention capacity decrease?

We visualize the attention capacity of our method and deep baselines on the ZINC dataset. The set of substructures is the same as the substructures used in our model, while the base vectors $E$ are defined as uniform distributions supported on corresponding substructures. Because attention capacity in Definition 4.1 depends on the norm of hidden representations, we normalize it by the norm of all the value vectors $\sum_{i=1...n} |h_i W^{VO}|_2$ to eliminate the influence of the representation norm. Note that for our model, only node tokens are considered. Additionally, because we explicitly tokenize substructures as tokens, we also directly compute the maximum difference between all the substructure tokens value vectors, $\max_{i,j \in \{1...m\}} |h_{i+n} W^{VO} - h_{j+n} W^{VO}|_2$, and normalize it by $\frac{1}{m} \sum_{i=1...m} |h_{i+n} W^{VO}|_2$, to illustrate the capacity of token representation of substructures. The results are plotted in Figure 1 (right).

First, the result of Graphormer and SAT reveals that attention capacity of substructures decreases in deep layers. Note that in the shallow layer the attention capacity of Graphormer increases, which is not contradicting our theory because we only claim that the upper bound of attention capacity decrease with depth. However, after the 24th layer, attention capacity decreases a lot, revealing the problem of deepening graph transformer. Note that because SAT uses GNN to encode substructures for query and key computation, it has a high attention capacity in the first layer. However, it decays fast due to the property of global attention.

Second, the result of our model indicates that local attention-based substructure tokens improve substructure representation capacity. The difference between substructure tokens is much larger

than the representation learned by attention on substructures, indicating learning substructure token representation by local attention can significantly improve the representation capacity of model, which supports our first motivation that introducing substructure-based local attention help model to encode substructures better.

Finally, compared to baselines, attention capacity of our model remains high across all the layers. Although it drops from 0.96 to 0.90 after the 24th layer, the dropping ratio is smaller than the baseline graph transformer. This supports our claim that substructure-based local attention helps alleviate attention capacity decrease. Note that the capacity of substructure tokens also decreases slower.

## H  PARAMETER NUMBER COMPARISON

We list parameter numbers of our model and baselines. Note that parameter numbers of Graphormer and SAT are computed by running their official code directly, and other baselines are from papers. The parameter number of SAT (*1) is different from reported in their papers, which may be due to code updates. The parameter number of our base model is comparable to previous works, and even less on CLUSTER. The parameter number of our deeper model is also comparable to deeper baselines, while our performance surpasses them.

|            | layer | PCQM4M-LSC  | ZINC      | CLUSTER   | PATTERN   |
|------------|-------|-------------|-----------|-----------|-----------|
| GCN        | -     | 2.0M        | 421k      | 571k      | 380k      |
| GIN        | -     | 3.8M        | 495k      | 684k      | 455k      |
| GT-Sparse  | -     | -           | 588,929   | 524,026   | 522,982   |
| GT-Full    | -     | -           | 588,929   | 524,026   | 522,982   |
| SAN-Sparse | -     | -           | 494,865   | 530,036   | 493,340   |
| SAN-Full   | -     | -           | 508,577   | 519,186   | 507,202   |
| Graphormer | x1    | 44,750,081  | 489,321   | -         | -         |
|            | x2    | 87,309,569  | 1,055,985 | -         | -         |
|            | x4    | 172,428,545 | 1,996,785 | -         | -         |
| SAT        | x1    | -           | 499,681   | 741,990   | 825,986   |
|            | x2    | -           | 991,873   | 1,480,806 | 1,646,978 |
|            | x4    | -           | 1,976,257 | 2,958,438 | 3,288,962 |
| Ours       | x1    | 45,563,393  | 612,705   | 382,865   | 612,705   |
|            | x2    | 88,122,881  | 1,078,225 | 686,225   | 1,083,105 |
|            | x4    | 173,241,857 | 2,019,025 | 989,585   | 2,023,905 |

## I  TIME COMPLEXITY

The time complexity of our methods is mainly due to the larger input size, which increases the cost of transformer. Given graph size $N$ and sampled substructure number $M$, the complexity of transformer is $O((M+N)^2)$. However, as we use sampling to reduce the substructure number, the substructure number is less than the graph size, i.e. $M < N$, so the complexity will not be more than four times of the standard transformer and remains $O(N^2)$.

## J  ABLATION STUDY ON NEIGHBOR SIZE

We conduct ablation studies for random walk neighbors on CLUSTER and PATTERN datasets. While we use size 10 neighbors in the paper, we compare the performance with sizes 5 and 15. The result is as follows:

|         | CLUSTER |      | PATTERN |      |
|---------|---------|------|---------|------|
|         | 12      | 48   | 12      | 48   |
| size 5  | 77.2    | 77.7 | 90.1    | 90.6 |
| size 15 | 77.1    | 78   | 89.6    | 90.4 |
| size 10 | 77.5    | 77.9 | 90      | 90.7 |

The result shows that the performance remains robust to different substructure sizes, especially the deeper model. Smaller substructures are easy for the model to learn its structure features, but they may not cover enough nodes and provide good graph structure information. Larger substructures are more informative about graph feature, but is more difficult for stable learning, especially for a shallow model.

