# OpenReview forum: "Are More Layers Beneficial to Graph Transformers?"
_ICLR.cc/2023/Conference — ICLR 2023 poster_

### Official Review · Reviewer_1tcv · 2022-10-22

**Confidence:** 3
**Correctness:** 3
**Technical Novelty And Significance:** 3
**Empirical Novelty And Significance:** 3
**Recommendation:** 6

**Clarity, Quality, Novelty And Reproducibility:**

The paper is clearly written and easy to follow.
The novelty of the proposed method looks significant.
There isn't much implementation detail provided so I feel it's nontrivial to reproduce the exact model.


**Strength And Weaknesses:**

Strengths:
- The theory proposed looks interesting, and the corresponding transformer structure introduced also look very novel.
- The authors conducted experiments on various benchmarks and the results look very competitive.
- The paper is clearly written and the motivation is easy to follow.

Weaknesses:
- Why there's such a negative correlation for Graph Transformer, while for Transformer in NLP the case is the larger, the better? Does the theoretical analysis also apply to Transformer in NLP?
- The proposed method has so many different designs so that it's hard to tell whether the proposed theory can justify what actually happened. Specifically, the authors proposed to tokenizing substructures subsampled from the graph. This is very different from the previous Graph Transformers as there's no sampling before.
- The authors didn't discuss the runtime overhead of the proposed method. Is it feasible to scale the proposed method on larger graphs?
- There isn't much ablation study in the paper. Specifically, the authors should study the size of substructure and how it affects the performance degradation with increased depth.

**Summary Of The Paper:**

In this paper, the authors first theoretically present the bottleneck of graph transformers’ performance with depth. The authors then propose a simple but effective substructure token based local attention mechanism in graph transformer, promoting focus on local substructure features of deeper graph transformer. Empirical results show that the proposed achieves state-of-the-art results on standard graph benchmarks with deeper models.

**Summary Of The Review:**

This paper provides interesting theoretical results and novel Graph Transformer design. The empirical results look good but not enough ablation studies. There's a nontrivial gap between the theory and the actual method proposed.

---

> ### Author Response · Authors · 2022-11-18
> **Response to Question 1 and 2**
>
> Response to question 1: Why there's such a negative correlation for Graph Transformer, while for Transformer in NLP the case is the larger, the better? Does the theoretical analysis also apply to Transformer in NLP?
>
> First, in other domains like NLP and CV, deep and large transformer are usually combined with large-scale pretraining, which improve their capacity and contributes to their performance a lot. For example, in the top layers of the pretrained transformer in CV, the attention map can distinguish semantic regions of objects in the image without supervision (Bao,2021), and similar results are found in NLP. Pretraining help deep model to allocate numerical problem and get stronger data inductive bias. In the graph field, pretraining has not been popularized as in NLP, and we believe this is also a promising direction for graph transformer studies.
>
> Second, there are also many works that reveal the problem in deepening transformer in NLP, while better initialization or architecture are proposed to solve optimization instability and over-smoothing, as presented in our related work. Compared to their works, our analysis focus on the attention capacity for substructures, specified for graph data.
>
> Third, compared to other domains, structure make up more part of graph data, which is the particularity of the graph field. And substructure is one of the basic intrinsic features of graph data, which is more complex than the case of sequential data structures like text and images. Thus it is not only more important but also more challenging for attention to focus on important substructures in graph data, so we mainly focus on the attention capacity of graph transformer.
>
>
> Finally, our method, including the substructure token and local attention, is also widely used in transformers of other fields. (1) Our method of tokenizing substructures closely corresponds to other fields. For nature language, sentences are separated into sub-words (Kudo,2018), which usually performs better than directly inputting characters. In computational vision, it's also found more beneficial to treat patches as tokens (Dosovitskiy,2020) than input image pixels. These works indicate the importance of proper tokenization of the input substructures. (2) Our method of local attention is also related to recent works in visual transformer, where local attention is shown to both improve performance and efficiency (Liu,2021), especially when training data is limited.
>
> Hangbo Bao, Li Dong, Songhao Piao, and Furu Wei. Beit: Bert pre-training of image transformers.
>
> Taku Kudo and John Richardson. Sentencepiece: A simple and language independent subword
> tokenizer and detokenizer for neural text processing.
>
> Alexey Dosovitskiy, Lucas Beyer, Alexander Kolesnikov, Dirk Weissenborn, Xiaohua Zhai, Thomas
> Unterthiner, Mostafa Dehghani, Matthias Minderer, Georg Heigold, Sylvain Gelly, et al. An im-
> age is worth 16x16 words: Transformers for image recognition at scale.
>
> Ze Liu, Yutong Lin, Yue Cao, Han Hu, Yixuan Wei, Zheng Zhang, Stephen Lin, and Baining Guo.
> Swin transformer: Hierarchical vision transformer using shifted windows.
>
>
>
>
>
>
> Response to question 2: The proposed method has so many different designs that it's hard to tell whether the proposed theory can justify what actually happened. Specifically, the authors proposed to tokenizing substructures subsampled from the graph. This is very different from the previous Graph Transformers as there's no sampling before.
>
> Our method contains mainly two parts: substructure tokens, and local attention on substructures. The main model is still a transformer without more modification. Substructure tokens do not influence the transformer architecture, but only help the model to find substructures; and local attention is a special case of global attention, where global attention is sparse and non-zero only on the substructure nodes.
>
> Our motivation for the proposed model is as follows. As Theorem 1 and 2 show, the decreasing attention capacity with depth reveals two problems in graph transformer. First, it is hard for graph transformer to pay attention to useful substructures when depth grows. Second, the feature learned by the attention mechanism loses expressiveness w.r.t substructures in deep layers. Both of these problems are pivotal to graph representation learning because the structural features of graphs are essential components of graph data. To address the first problem, it's natural to introduce substructure inductive bias into graph transformer to make up for the deficiency of feature selection. Theorem 3 mainly stresses the second problem, showing that introducing substructures into attention also improves the sensitivity of model to these features.

---

> ### Author Response · Authors · 2022-11-18
> **Response to Question 3 and 4**
>
> Response to question 3: The authors didn't discuss the runtime overhead of the proposed method. Is it feasible to scale the proposed method on larger graphs?
>
>
>
> The time complexity of our methods is mainly due to the larger input size, which increases the cost of transformer. Given graph size $N$ and sampled substructure number $M$, the complexity of transformer is $O((M+N)^2)$. However, as we use sampling to reduce the substructure number, the substructure number is less than the graph size, i.e. $M<N$,  so the complexity will not be more than four times of the standard transformer and remains $O(N^2)$.
>
>
>
> Response to question 4: There isn't much ablation study in the paper. Specifically, the authors should study the size of the substructure and how it affects the performance degradation with increased depth.
>
> Thank you for your constructive suggestions! In fact, for geometric substructures, we use various sizes of different substructures, covering the different sizes of substructures. For example, circle size varies from 3 to 8, star from 2 to 6, and path from 4 to 8. As for the neighborhoods substructure, 2-hop substructure is most suitable for ZINC and PCQM4M-LSC, because 1-hop substructure is trivial, and more than 2 hops substructures will contain too many nodes. So we conduct ablation studies for random walk neighbors on CLUSTER and PATTERN datasets. While we use size 10 neighbors in the paper, we compare the performance with sizes 5 and 15. The result is as follows:
>
> | ~       | CLUSTER | ~    | PATTERN |      |
> |---------|---------|------|---------|------|
> | ~       | 12      | 48   | 12      | 48   |
> | size 5  | 77.2    | 77.7 | 90.1    | 90.6 |
> | size 15 | 77.1    | 78   | 89.6    | 90.4 |
> | size 10 | 77.5    | 77.9 | 90      | 90.7 |
>
>
>
> The result shows that the performance remains robust to different substructure sizes, especially the deeper model. Smaller substructures are easy for the model to learn its structure features, but they may not cover enough nodes and provide good graph structure information. Larger substructures are more informative about graph feature, but is more difficult for stable learning, especially for a shallow model. We add this experiment in the appendix of the paper due to the space limitation.

---

### Official Review · Reviewer_KU4L · 2022-10-23

**Confidence:** 2
**Correctness:** 3
**Technical Novelty And Significance:** 3
**Empirical Novelty And Significance:** 3
**Recommendation:** 6

**Clarity, Quality, Novelty And Reproducibility:**

The work is novel to the best of my knowledge and seems of high quality. The paper is clearly written mostly, with small confusing bits potentially due to typos. Specifically, in the definition of M_{ij} under equation 10, should it be i \in {...} instead of i+n \in {...}? Or is there some special relation between node i and substructure i + n?


**Strength And Weaknesses:**

Strength
- Depth limitation of gnns and graph transformers has been a key issue in graph ML. The theoretical analysis reveals insights for solving this.
- The proposed method is conceptually simple and motivated by theoretical results.
- The proposed method is tested over a diverse set of datasets and tasks and demonstrates empirical gains.

Weaknesses
- The performance improvements with depth seem pretty small relative to the variances on most datasets.
- The method seems not completely correspond to the theoretical motivation: in theorem 3 substructure based local attention is defined as  "where each node only attends nodes that belong to the same substructures". If I understand correctly, in DeepGraph each node can still attend to all other nodes, but only that the substructure token only attends to and by the nodes in the substructure. Can this be further justified?

**Summary Of The Paper:**

This paper shows the theoretical bottleneck on attention capacity for transformer depth scaling and proposed a new graph transformer model called DeepGraph to address it. The proposed architecture relies on substructure-based attention mechanism, where there are extra substructure tokens that correspond to sampled substructures. Specifically, each substructure token only attends to tokens within it. The model is shown to increase performance on several datasets as depth increases.

**Summary Of The Review:**

Overall, the paper is addressing an important issue. The theoretical results and proposed methods seem sound and well tested. The concern is about the significance of the experimental results on depth improvement.

---

> ### Author Response · Authors · 2022-11-18
> **Response to Reviewer**
>
> \section{Reviewer3}
>
> Response to question 1: The performance improvements with depth seem pretty small relative to the variances on most datasets.
>
> Our model achieves 0.02 MAE improvement on ZINC dataset, compared to the previous state-of-art 0.094, and improves accuracy on PATTERN from 86.87 to 90.66, which are both considerable improvements considering the high performance of the previous method, relative to the variances. We also surpass previous SOTA results on PCQM4M, while achieving comparable performance on CLUSTER relative to the variances.
>
> Response to question 2: The method seems not completely correspond to the theoretical motivation: in theorem 3 substructure-based local attention is defined as "where each node only attends nodes that belong to the same substructures". If I understand correctly, in DeepGraph each node can still attend to all other nodes, but only the substructure token only attends to and by the nodes in the substructure. Can this be further justified?
>
> That's exactly how our model is designed. Our motivation is as follows:
>
> As Theorem 1 and 2 show, the decreasing attention capacity with depth reveals two problems in graph transformer. First, it is hard for a graph transformer to pay attention to useful substructures when depth grows. Second, the feature learned by the attention mechanism loses expressiveness w.r.t substructures in deep layers. Both of these problems are pivotal to graph representation learning because the structural features of graphs are essential components of graph data. To address the first problem, it's natural to introduce substructure inductive bias into graph transformer to make up for the deficiency of feature selection. Theorem 3 mainly stresses the second problem, showing that introducing substructures into attention also improves the sensitivity of the model to these features. We revised paragraphs about our motivation after theorem 2.
>
> However, replacing global attention directly with local attention will lose the main benefit of transformer for long-range dependencies. To achieve a better global-local encoding balance, we propose to first add substructure information by tokenizing substructures, which helps model to find substructures, and then apply partial local attention in this situation, where local attention is applied between nodes and substructure tokens while the original global attention between nodes is preserved. This has been stated at the beginning of Section 5 Approach.
>
> Response to questions in Clarity, Quality, Novelty And Reproducibility: In the definition of $M_{ij}$ under equation 10, should it be $i \in$ {...} instead of $i+n \in$ {...}?
>
> Here $i+n$ is just the token id of substructure $i$, because we put them with nodes together. Our input embedding contains nodes tokens embedding $\{h_1,h_2, \dots ,h_n\}$ and substructure tokens embedding $\{h_{n+1},\dots,h_{n+m}\}$ corresponding to substructures $\{G^S_1, G^S_2 \dots G^S_m\}$. We use mask to make substructure token only attend to and by the nodes in the substructure, so the mask value is $M_{ij}=-\infty$ if $i+n \in \{n+1,n+2,\dots n+m\}, j \not\in N^S_{i}$ or inverse. $i+n \in \{n+1,n+2,\dots n+m\}$ means token $i+n$ is substructure token of substructure $G^S_i$, so the condition is $j \not\in N^S_{i}$.

---

### Official Review · Reviewer_n5aK · 2022-10-25

**Confidence:** 4
**Correctness:** 2
**Technical Novelty And Significance:** 2
**Empirical Novelty And Significance:** 2
**Recommendation:** 5

**Clarity, Quality, Novelty And Reproducibility:**

Clarity can be improved. Quality is limited by novelty in model design. However if the theoretical analysis is correct, the novelty can be improved. Please see the comments about Definition 1 and Theorem 1,2.

**Strength And Weaknesses:**

**Strength**:
1. Substructure based GNNs are widely explored last year and also influences the graph transformer area. It's interesting to see that adding substructure directly helps improving current graph transformers.
2. The author proposes a measure to define the attention capacity for substructures. This is an interesting angle.
3. The improvement over real-world dataset is great, especially on PATTERN dataset.

**Weakness**:
1. Notation and writing is not clear and is too hard to follow. I suggest the author to revise the paper extensively to make the notation easier to follow. For example, the author use $G^S$ for general notation of subgraph, which is really confusing as I was originally thinking that $G^S = G[S]$ representing the set S induced subgraph on G.  For general subgraph pattern, I suggest the author remove the G notation, just use something like $S$ directly, as there is no relation to G. The section 4 needs to be greatly improved. For example, even the first paragraph inside 4.1 is hard to follow. All notations of graph, subgraph, pattern, and support are too hard to follow. Also $supp(e) \subseteq  N^S$ instead of $supp(e) \in  N^S$. There are so many confusing notations and I cannot read and evaluate the section 4. Another example is the $\triangle$ notation...it represents easy stuff in principle but the notation makes the easy stuff hard to interpret. Again, like $Attn_{A}$,  the underscript is never mentioned before using.
2. Definition 1 of measuring attention capacity is questionable. There are learnable W inside and it's not possible to analyze the F norm directly, as W can have any scale to enlarge the F norm. Also, there is no relationship between small F norm and substructure indistinguishability. (a-b small doesn't imply a=b for graph separation case) I suggest the author at least give some empirical example to link the small F norm and the bad ability of identifying substructure.
3. Theorem 1 and Theorem 2 I cannot read yet, after the author revise the section 4 I would like to check the theorem correctness.
4. Section 5 substructure sampling is very similar to the one designed in [Zhao et al. ICLR 22]. The author should mention with reference.
5. The designed method is a special case of [Kim et al. NeurIPS 22], and it doesn't provide much new contribution to model design.
6. The ZINC experiment may need 500K parameter restriction in comparison.

[Kim et al. NeurIPS 22] Pure transformer are powerful graph learners.
[Zhao et al. ICLR 22] From stars to subgraphs



**Summary Of The Paper:**

The author studies the effect of number of layers in graph transformer model, and claims that the previously used graph transformer layer has decreasing attention capacity on attending substructures with increasing attention layers. Then the author improves graph transformer with adding sampled substructures as token along with nodes. Extensive experiments over large real-world datasets are conducted to show its improvement.

**Summary Of The Review:**

The author added substructure tokens to current graph transformer and show empirically improvement. In the meantime, the author try to answer why increasing depth doesn't benefit the performance for previous graph transformers, with a measure of attention capacity. However the relationship between the defined attention capacity and the empirical performance drop of increasing depth is not clear.

---

> ### Author Response · Authors · 2022-11-18
> **Response to Question 5,1 and 4**
>
> Response to Question 5: The designed method is a special case of [Kim et al. NeurIPS 22], and it doesn't provide much new contribution to model design.
>
> [Kim et al. NeurIPS 22], which we also referred to in related work, proposed TokenGT which treats all nodes and edges as independent tokens, augments them with absolute position encoding and token type encodings, and feed them to a Transformer. Compared to their works, our method proposes to tokenize general substructures subsampled from the graph and apply local attention to related nodes within substructures.
>
> **In our view, our method is remarkably different from TokenGT in three aspects**: First, we propose to tokenize general substructures rather than original nodes and edges only, which is a more general case of graph tokenization. Second, we also propose local attention on substructures, which helps to introduce substructure inductive bias and allocate attention capacity vanishing. Finally, the motivation of our work is different from TokenGT. TokenGT proposes edge tokens with absolute position embedding for theoretical expressive power, while our works focus on the attention capacity for substructure features.
>
> Response to Question 1:
>
> Thanks for your suggestion on writing! We revised our notation, especially in Subsection 4.1. We simplified the notation of $\Delta$, correct the usage of $\operatorname{supp}$, and reorganized the order of different definitions to make it easier to follow.
>
> Q: The author use  for general notation of subgraph, which is really confusing as I was originally thinking that  representing the set S induced subgraph on G.
>
> A: We leave the notation of $G^S$ because the substructures are indeed induced subgraphs on G in this work. This is because the backbone graph transformer only takes nodes as tokens, and attention is only applied to nodes of the graph, so it is unable to tell whether arbitrary subgraphs or induced subgraph is focused by attention. In other words, attention to arbitrary subgraphs is not well-defined. So in our algorithm, only induced subgraphs are used for encoding. We noted this in our Preliminary ($N^S \subset N, E^S=(N^S \times N^S) \cap E$, i.e. nodes of $G^S$ form a subset of the graph $G$ and edges are all the existing edges in $G$ between nodes subset), but without additional clarify that substructures we consider are induced only. We add this in the new version. Furthermore, as far as we know, the notation $G^S$ is commonly used both for arbitrary subgraph and induced subgraph, for example in Chen,2020 and Bouritsas,2022. The $G$ notation is used to stress the corresponding between substructure and the graph $G$, because the node features and edge features are inherited from the original graph $G$.
>
> Zhengdao Chen, Lei Chen, Soledad Villar, and Joan Bruna. Can graph neural networks count
> substructures?
>
> Giorgos Bouritsas, Fabrizio Frasca, Stefanos P Zafeiriou, and Michael Bronstein. Improving graph
> neural network expressivity via subgraph isomorphism counting
>
> Response to Question 4: Section 5 substructure sampling is very similar to the one designed in [Zhao et al. ICLR 22]. The author should mention with reference.
>
> Thanks for the suggestions! We indeed get inspired by the thought of subsampling for uniform cover, which we originally referred to in related work, where the greedy minimum set cover algorithm is applied to iteratively select the subgraph containing the maximum number of uncovered nodes. However, the greedy minimum set cover algorithm lacks randomness, and is inefficient when the number of candidate substructures is huge. Our algorithm is designed to use randomized top-k sampling on a randomly selected subset of substructures to speed up sampling and yield more variant sampling results. We will mention the reference in the method section.

---

> ### Author Response · Authors · 2022-11-18
> **Response to Question 2 (part 1)**
>
> Response to Question 2:
>
> Q: Definition 1 of measuring attention capacity is questionable. There are learnable W inside and it's not possible to analyze the F norm directly, as W can have any scale to enlarge the F norm.
>
>
> A: Theoretically, the scale of $W_l^{VO}$ does not change the relative scale between attention capacity and hidden representations norm, which is the intrinsic problem that matters. $W_l^{VO}$ indeed influences both attention capacity and the norm of the learned hidden representations with the same scale (due to the property of ReLU function), but it does not help to discriminate different attention patterns on substructures if the learned hidden representation and attention capacity scale the same factor.
>
>
> In practice, we deal with this issue by considering the fact that, properly designed initialization like Xavier uniform (Glorot,2010), and layer normalization are used to keep the representation norm the same across layers, which helps for stable learning. Thus we assume the representation norm is the same in each layer, as stated in Theorem 1, due to the properly designed initialization which keeps the output norm the same as the input, and layer normalization that is used in transformer. Thus $W_l^{VO}$ only appears once in Theorem 1, which is the parameter of layer $l$. While we make this assumption just for convenience, it's also feasible to explicitly product $\prod_{i=1}^{l-1} |W_i^{VO}|_2$ in the upper bound of attention capacity, but this factor will also appear in the norm of hidden representation, which does not matter if we consider their relative scale.
>
> Xavier Glorot and Yoshua Bengio. Understanding the difficulty of training deep feedforward neural
> networks.

---

> ### Author Response · Authors · 2022-11-18
> **Response to Question 2 (part 2)**
>
> Q: Also, there is no relationship between a small F norm and substructure indistinguishability. (a-b small doesn't imply a=b for graph separation case)
>
> A: Indeed, a small F norm doesn't mean that model can not absolutely distinguish different substructures, but it will be harder for the model to distinguish different substructures and learn to focus on important substructures. It has commonly been assumed that feature similarity influence the difficulty of discrimination, for example in previous work about over-smoothing of graph neural network (Oono,2019), where the similarity between node representation is considered harmful for node tasks.
>
> Kenta Oono and Taiji Suzuki. Graph neural networks exponentially lose expressive power for node
> classification.
>
> Q: I suggest the author at least give some empirical examples to link the small F norm and the bad ability to identify substructure. (Also Answer to Question in Summary: the relationship between the defined attention capacity and the empirical performance drop of increasing depth is not clear.)
>
> A: Thanks for the suggestions! The importance of substructures is stated in related work (Graph substructure). To better illustrate that larger attention capacity on substructures is strongly correlated with better performance, we empirically explore the capacity of the model for the substructure, and validate our theoretical results. We aim to answer the following three questions: (1) Does attention capacity really decrease in deeper graph transformer? (2) Can our method, i.e., local attention with substructure tokens, help to increase the model capacity of representing substructures? (3) Can our method help to alleviate attention capacity decrease? Note that the last two questions also illustrate the relationship between attention capacity and model performance, because it has been shown that our method empirically improve model performance.
>
>  We compare the attention capacity on substructures of our method and Graphormer on ZINC dataset. The substructures set is the same as substructures used in our model for fair comparison, while the base vectors $E$ are defined as uniform distribution supported on corresponding substructures in computation.
>
>  We compute the attention capacity according to Definition 1 for a single output position,
>
>  $\max_{c_1,c_2 \in \{c| c \in [0,1]^{m},c^T \textbf{1}=1\} } |c_1^TE^THW^{VO}-c_2^TE^THW^{VO}|_2$,
>
>  and normalize it by the mean of l2 norm of all the value vector of nodes
>
>  $\frac{1}{n}\sum_{i=1 \dots n}|h_iW^{VO}|_{2}$,
>
>  to eliminate the influence of the representation norm. Note that for our model, only node tokens are considered. Additionally, because we explicitly tokenize substructures as tokens, we also directly compute the maximum difference between all the substructure tokens value vectors,
>
>  $\max_{i,j \in \{1 \dots m\}}|h_{i+n}W^{VO}-h_{j+n}W^{VO}|_2$,
>
>  and normalize it by
>
>  $\frac{1}{m}\sum_{i=1 \dots m}|h_{i+n}W^{VO}|_{2}$,
>  to illustrate the capacity of token representation of substructures. We report the capacity of different layers for comparison. The result of each layer is plotted in Appendix. The results of 12th, 24th, 36th and  48th layer are as follows:
>
>
> | Layer          | 12     | 24     | 36     | 48     |
> |----------------|--------|--------|--------|--------|
> | Graphormer     | 0.9348 | 0.9837 | 0.9121 | 0.6719 |
> | Ours           | 0.9170 | 0.9648 | 0.9230 | 0.9005 |
> | Ours Sub Token | 1.3453 | 1.3598 | 1.3529 | 1.2862 |
>
> First, the result of Graphormer reveals that attention capacity of substructures decreases in deep layers. Note that in the shallow layer the attention capacity increases, which is not contradicting our theory because we only claim that the upper bound of attention capacity decrease with depth. However, after the 24th layer, attention capacity decreases a lot, revealing the problem of deepening graph transformer.
>
> Second, the result of the third row indicates that local attention-based substructure tokens improve substructure representation capacity. The difference between substructure tokens is much larger than the representation learned by attention on substructures, indicating learning substructure token representation by local attention can significantly improve the representation capacity of model.
>
> Finally, compared to Graphormer, attention capacity of our model remains high across all the layers. Although it drops from 0.96 to 0.90 after the 24th layer, the dropping ratio is smaller than the vanilla graph transformer. This supports our claim that substructure-based local attention helps alleviate attention capacity decrease. Note that capacity of substructure tokens also decreases slower.
>
> The last two conclusion reveal that our method helps to increase representation capacity on substructures, which also improve model performance. We hope this help to support our motivation.

---

> ### Author Response · Authors · 2022-11-18
> **Response to Question 6**
>
> Response to Question 6: The ZINC experiment may need 500K parameter restriction in comparison.
>
> Thanks for the constructive suggestions! We list the parameter numbers of our model and baselines. Note that parameter numbers of Graphormer and SAT are computed by running their official code directly, and other baselines are from papers. The parameter number of SAT (*1) is different from reported in their papers, which may be due to code updates. The parameter number of our base model is comparable to previous works, and even less on CLUSTER. The parameter number of our deeper model is also comparable to deeper baselines, while our performance surpasses them, as shown in Subsection 6.5. We add this in Appendix due to space limitations.
>
>
> | ~          | layer | PCQM4M-LSC  | ZINC      | CLUSTER   | PATTERN   |
> |------------|-------|-------------|-----------|-----------|-----------|
> | GCN        | -     | 2.0M        | 421k      | 571k      | 380k      |
> | GIN        | -     | 3.8M        | 495k      | 684k      | 455k      |
> | GT-Sparse  | -     | -           | 588,929   | 524,026   | 522,982   |
> | GT-Full    | -     | -           | 588,929   | 524,026   | 522,982   |
> | SAN-Sparse | -     | -           | 494,865   | 530,036   | 493,340   |
> | SAN-Full   | -     | -           | 508,577   | 519,186   | 507,202   |
> | Graphormer | x1    | 44,750,081  | 489,321   | -         | -         |
> | ~          | x2    | 87,309,569  | 1,055,985 | -         | -         |
> | ~          | x4    | 172,428,545 | 1,996,785 | -         | -         |
> | SAT        | x1    | -           | 499,681   | 741,990   | 825,986   |
> | ~          | x2    | -           | 991,873   | 1,480,806 | 1,646,978 |
> | ~          | x4    | -           | 1,976,257 | 2,958,438 | 3,288,962 |
> | Ours       | x1    | 45,563,393  | 612,705   | 382,865   | 612,705   |
> | ~          | x2    | 88,122,881  | 1,078,225 | 686,225   | 1,083,105 |
> | ~          | x4    | 173,241,857 | 2,019,025 | 989,585   | 2,023,905 |

---

### Official Review · Reviewer_n5Mh · 2022-11-01

**Confidence:** 3
**Correctness:** 3
**Technical Novelty And Significance:** 3
**Empirical Novelty And Significance:** 2
**Recommendation:** 6

**Clarity, Quality, Novelty And Reproducibility:**

The paper is generally quite clear, and the empirical studies is relatively extensive. I did not check the math carefully but I did go through all of the proof. The paper provides a good insight to an important problem that exists in graph transformers, and the empirical results seem solid. However, structural encodings is not a completely new idea. The authors mentioned that the code will be provided.

**Strength And Weaknesses:**

Strengths:
1) The paper tries to provide both theoretical and empirical insights on the importance of substructure sensitivity to the capacity of graph transformers.
2) The writing is generally clear.
3) I think the problem in itself is interesting (i.e., how can we train deeper graph transformers).
4) Good empirical results and ablative studies.

Weaknesses/questions:
1) While it's nice that the paper provides theoretical analysis on the model capacity, it is unclear to me how meaningful they are. Specifically, for example:
    - Inequality (6) provides an upper bound. In practice, $C_1$ and $C_2$ also depends on $H_\ell$, so they are not independent. This makes the inequality (12) in the Appendix potentially vacuous. Moreover, while $\alpha_i$ is accumulated in inequality (6) across layers indeed, it is unclear how $W_\ell^{VO}$ (which is learnable) evolves across depth. Could it balance this decay process? As another example, Theorem 3 is merely comparing two upper bounds. Without any evidence on the looseness of these upper bounds, it's hard to evaluate the value of this theorem. It'd be great if the authors could elaborate on issues like these.
    - The theoretical results applies mainly to the input level. $G_1^S, \dots, G_m^S$ makes sense at the input layer where each node contains the information only about itself. But starting from layer 2, the hidden units already combines information from all lower-level nodes, making the substructure argument a bit vague.
2) Given the assumptions, the theoretical results should also largely apply to the original (non-graph) transformers, such as those used in NLP. And yet we are able to train very deep transformers there, even though substructures do exist in language. Could the authors discuss more what their results imply in non-graph applications?
3) Do techniques that allow very deep training of conventional transformers help deepen graph transformers (e.g., ReZero)? This would be an interesting ablative study that demonstrates why the substructure attention is a better alternative for graphs.
4) For PCQM4M-LSC and ZINC, the # of nodes in these tasks is usually very small (e.g., <60). I'm a bit surprised that substructure is still very useful in these cases as global attention is not operating on a huge graph anyway.
5) Could the authors provide the # of parameters for each model in Table 1?

**Summary Of The Paper:**

The paper studies the limitations to deepen graph transformers and argue that substructure learning gets increasingly harder in the canonical formulation of the graph transformers. To address this limitation, the paper proposes a variant of graph transformers that explicitly models substructure attention and encoding. Empirical results suggest that the proposed DeepGraph approach is effective and competitive.

**Summary Of The Review:**

Overall, I find the paper interesting and it provides a good insight into how substructure modeling plays a role in the attention and expressivity of modern graph transformers. There is still an obvious gap between the theory section and the empirical section (e.g., the paper did not empirically verify any of the theoretical conclusions), but the empirical results are quite good. I'm inclined to acceptance on the condition that my questions can be answered satisfactorily.

---

> ### Author Response · Authors · 2022-11-18
> **Response to Question 1 (part 1)**
>
>
> Response to Question 1:
>
>
> Q: $C_1$ and $C_2$ also depends on $H_l$, so they are not independent:
>
> A: The inequality (12) in the Appendix can be upper bounded by matrix inequation, where the equation holds if columns of $E^THW^{VO}$ are the first right singular vector of $(C_1-C_2)^T$. For example, if each row of $(C_1-C_2)^T$ is $(1,-1,0,0)$, and a column of $E^THW^{VO}$ is $(a,-a,0,0)^T$, then $(C_1-C_2)^T((a,-a,0,0)^T)$ is $(2a,2a,2a,2a)^T$, the norm of which is increased by a constant factor $|(C_1-C_2)^T|_2=\sqrt{2m}$. Our original analysis splits $(C_1-C_2)^T$ here, and gets a bound $\sqrt{2}$ larger than the tightest bound, which is indeed sub-optimal. The tightest bound should be $\sqrt{2m}$. We revised the coefficient here, and thank you for suggestions!
>
> Q: It is unclear how $W_l^{VO}$ (which is learnable) evolves across depth:
>
>
>
>
>
> A: Theoretically, the scale of $W_l^{VO}$ does not change the relative scale between attention capacity and hidden representations norm, which is the intrinsic problem that matters. $W_l^{VO}$ indeed influences both attention capacity and the norm of the learned hidden representations with the same scale (due to the property of ReLU function), but it does not help to discriminate different attention patterns on substructures if the learned hidden representation and attention capacity scale the same factor.
>
>
> In practice, we deal with this issue by considering the fact that, properly designed initialization like Xavier uniform (Glorot,2010), and layer normalization are used to keep the representation norm the same across layers, which helps for stable learning. Thus we assume the representation norm is the same in each layer, as stated in Theorem 1, due to the properly designed initialization which keeps the output norm the same as the input, and layer normalization that is used in transformer. Thus $W_l^{VO}$ only appears once in Theorem 1, which is the parameter of layer $l$. While we make this assumption just for convenience, it's also feasible to explicitly product $\prod_{i=1}^{l-1} |W_i^{VO}|_2$ in the upper bound of attention capacity, but this factor will also appear in the norm of hidden representation, which does not matter if we consider their relative scale.
>
> Xavier Glorot and Yoshua Bengio. Understanding the difficulty of training deep feedforward neural
> networks.

---

> ### Author Response · Authors · 2022-11-18
> **Response to Question 1 (part 2)**
>
> Q: Theorem 3 is merely comparing two upper bounds.  (The paper did not empirically verify any of the theoretical conclusions)
>
> A: In theorem 3 we compare the upper bounds, similar to Rong,2019, to illustrate how local attention on substructures can alleviate the worst case of attention capacity decay.
>
>
>  We empirically explore the capacity of the model for the substructure, and validate our theoretical results. We aim to answer the following three questions: (1) Do attention capacity really decreases in deeper graph transformer? (2) Can our method, i.e., local attention with substructure tokens, help to increase the model capacity of representing substructures? (3) Can our method, i.e., local attention with substructure tokens, help to alleviate attention capacity decrease?
>
>
>
>  We compare the attention capacity on substructures of our method and Graphormer on ZINC dataset. The substructures set is the same as substructures used in our model for fair comparison, while the base vectors $E$ are defined as uniform distribution supported on corresponding substructures in computation.
>
>  We compute the attention capacity according to Definition 1 for a single output position,
>
>  $\max_{c_1,c_2 \in \{c| c \in [0,1]^{m},c^T \textbf{1}=1\} } |c_1^TE^THW^{VO}-c_2^TE^THW^{VO}|_2$,
>
>  and normalize it by the mean of l2 norm of all the value vector of nodes
>
>  $\frac{1}{n}\sum_{i=1 \dots n}|h_iW^{VO}|_{2}$,
>
>  to eliminate the influence of the representation norm. Note that for our model, only node tokens are considered. Additionally, because we explicitly tokenize substructures as tokens, we also directly compute the maximum difference between all the substructure tokens value vectors,
>
>  $\max_{i,j \in \{1 \dots m\}}|h_{i+n}W^{VO}-h_{j+n}W^{VO}|_2$,
>
>  and normalize it by
>
>  $\frac{1}{m}\sum_{i=1 \dots m}|h_{i+n}W^{VO}|_{2}$,
>
>  to illustrate the capacity of token representation of substructures. We report the capacity of different layers for comparison. The result of each layer is plotted in Appendix. The results of 12th, 24th, 36th and  48th layer are as follows:
>
> | Layer          | 12     | 24     | 36     | 48     |
> |----------------|--------|--------|--------|--------|
> | Graphormer     | 0.9348 | 0.9837 | 0.9121 | 0.6719 |
> | Ours           | 0.9170 | 0.9648 | 0.9230 | 0.9005 |
> | Ours Sub Token | 1.3453 | 1.3598 | 1.3529 | 1.2862 |
>
>
> First, the result of Graphormer reveals that the attention capacity of substructures decreases in deep layers. Note that in the shallow layer the attention capacity increases, which is not contradicting our theory because we only claim that the upper bound of attention capacity decrease with depth. However, after the 24th layer, attention capacity decreases a lot, revealing the problem of deepening graph transformer.
>
> Second, the result of the third row indicates that local attention-based substructure tokens improve substructure representation capacity. The difference between substructure tokens is much larger than the representation learned by attention on substructures, indicating learning substructure token representation by local attention can significantly improve the representation capacity of model, which supports our first motivation that introducing substructure-based local attention help model to encode substructures better.
>
> Finally, compared to Graphormer, attention capacity of our model remains high across all the layers. Although it drops from 0.96 to 0.90 after the 24th layer, the dropping ratio is smaller than the vanilla graph transformer. This supports our claim that substructure-based local attention helps alleviate attention capacity decrease. Note that the capacity of substructure tokens also decreases slower.
>
> Yu Rong, Wenbing Huang, Tingyang Xu, and Junzhou Huang. Dropedge: Towards deep graph
> convolutional networks on node classification.
>
>
> Q: Starting from layer 2, the hidden units already combine information from all lower-level nodes, making the substructure argument a bit vague.
>
> A: While substructure is one of the basic inductive bias of graph data, many works show that models still need the inductive bias of data in deep layers. For deep graph neural networks, graph-inductive-biased GNN layers are stacked many times, even if each node also is informative to the whole graph when the number of layers is larger than the graph diameter, and similar for deep convolutional networks. Another example is the transformer in CV, where in top layers, the attention map still distinguishes semantic regions of objects in the image (Bao,2021). Finally, relative position encoding is applied to each layer of the graph transformer, which still highlights the inductive bias of the structure used in deep layers. These examples suggest that applying inductive bias of substructure in deep layers may be still beneficial.
>
> Hangbo Bao, Li Dong, Songhao Piao, and Furu Wei. Beit: Bert pre-training of image transformers.

---

> ### Author Response · Authors · 2022-11-18
> **Response to Question 2, 3 and 4**
>
> Response to Question 2: Could the authors discuss more what their results imply in non-graph applications?
>
> First, in other domains like NLP and CV, deep and large transformer are usually combined with large-scale pretraining, which improve their capacity and contributes to their performance a lot. For example, in the top layers of the pretrained transformer in CV, the attention map can distinguish semantic regions of objects in the image without supervision (Bao,2021), and similar results are found in NLP. Pretraining help deep model to allocate numerical problem and get stronger data inductive bias. In the graph field, pretraining has not been popularized as in NLP, and we believe this is also a promising direction for graph transformer studies.
>
> Second, there are also many works that reveal the problem in deepening transformer in NLP, while better initialization or architecture are proposed to solve optimization instability and over-smoothing, as presented in our related work. Compared to their works, our analysis focus on the attention capacity for substructures, specified for graph data.
>
> Third, compared to other domains, structure make up more part of graph data, which is the particularity of the graph field. And substructure is one of the basic intrinsic features of graph data, which is more complex than the case of sequential data structures like text and images. Thus it is not only more important but also more challenging for attention to focus on important substructures in graph data, so we mainly focus on the attention capacity of graph transformer.
>
>
> Finally, our method, including the substructure token and local attention, is also widely used in transformers of other fields. (1) Our method of tokenizing substructures closely corresponds to other fields. For nature language, sentences are separated into sub-words (Kudo,2018), which usually performs better than directly inputting characters. In computational vision, it's also found more beneficial to treat patches as tokens (Dosovitskiy,2020) than input image pixels. These works indicate the importance of proper tokenization of the input substructures. (2) Our method of local attention is also related to recent works in visual transformer, where local attention is shown to both improve performance and efficiency (Liu,2021), especially when training data is limited.
>
> Hangbo Bao, Li Dong, Songhao Piao, and Furu Wei. Beit: Bert pre-training of image transformers.
>
> Taku Kudo and John Richardson. Sentencepiece: A simple and language independent subword
> tokenizer and detokenizer for neural text processing.
>
> Alexey Dosovitskiy, Lucas Beyer, Alexander Kolesnikov, Dirk Weissenborn, Xiaohua Zhai, Thomas
> Unterthiner, Mostafa Dehghani, Matthias Minderer, Georg Heigold, Sylvain Gelly, et al. An im-
> age is worth 16x16 words: Transformers for image recognition at scale.
>
> Ze Liu, Yutong Lin, Yue Cao, Han Hu, Yixuan Wei, Zheng Zhang, Stephen Lin, and Baining Guo.
> Swin transformer: Hierarchical vision transformer using shifted windows.
>
>
> Response to Question 3: Do techniques that allow very deep training of conventional transformers help deepen graph transformers (e.g., ReZero)?
>
> We mentioned Rezero in related work, and we adopt and compare it to the more recent method Deepnorm which is also technique stabilizing transformer optimization. We study the effect of Deepnorm in the ablation study, showing that Deepnorm indeed helps train our deep model to some degree. In section 6.5, we also compare with the deepened versions of the state-of-arts augmented by recent algorithms designed for training deep transformers, including the fusion method (Shi,2021), and reattention method (Zhou,2021).
>
> Han Shi, JIAHUI GAO, Hang Xu, Xiaodan Liang, Zhenguo Li, Lingpeng Kong, Stephen MS Lee,
> and James Kwok. Revisiting over-smoothing in bert from the perspective of graph.
>
> Daquan Zhou, Bingyi Kang, Xiaojie Jin, Linjie Yang, Xiaochen Lian, Zihang Jiang, Qibin Hou,
> and Jiashi Feng. Deepvit: Towards deeper vision transformer.
>
>
>
> Response to Question 4: I'm a bit surprised that substructure is still very useful in these cases as global attention is not operating on a huge graph anyway.
>
> Although molecular graphs are not very large, for example on average 23.16 nodes in ZINC dataset, the possible graph data space is quite huge, so it's still nontrivial for the model to learn important substructures from a molecular graph. So adding substructure inductive bias may be a good solution to make up for this weakness.

---

> ### Author Response · Authors · 2022-11-18
> **Response to Question 5**
>
> Response to Question 5: Could the authors provide the number of parameters for each model in Table 1?
>
> Thanks for the constructive suggestions! We list the parameter numbers of our model and baselines. Note that parameter numbers of Graphormer and SAT are computed by running their official code directly, and other baselines are from papers. The parameter number of SAT (*1) is different from reported in their papers, which may be due to code updates. The parameter number of our base model is comparable to previous works, and even less on CLUSTER. The parameter number of our deeper model is also comparable to deeper baselines, while our performance surpasses them, as shown in Subsection 6.5. We add this in Appendix due to space limitations.
>
> | ~          | layer | PCQM4M-LSC  | ZINC      | CLUSTER   | PATTERN   |
> |------------|-------|-------------|-----------|-----------|-----------|
> | GCN        | -     | 2.0M        | 421k      | 571k      | 380k      |
> | GIN        | -     | 3.8M        | 495k      | 684k      | 455k      |
> | GT-Sparse  | -     | -           | 588,929   | 524,026   | 522,982   |
> | GT-Full    | -     | -           | 588,929   | 524,026   | 522,982   |
> | SAN-Sparse | -     | -           | 494,865   | 530,036   | 493,340   |
> | SAN-Full   | -     | -           | 508,577   | 519,186   | 507,202   |
> | Graphormer | x1    | 44,750,081  | 489,321   | -         | -         |
> | ~          | x2    | 87,309,569  | 1,055,985 | -         | -         |
> | ~          | x4    | 172,428,545 | 1,996,785 | -         | -         |
> | SAT        | x1    | -           | 499,681   | 741,990   | 825,986   |
> | ~          | x2    | -           | 991,873   | 1,480,806 | 1,646,978 |
> | ~          | x4    | -           | 1,976,257 | 2,958,438 | 3,288,962 |
> | Ours       | x1    | 45,563,393  | 612,705   | 382,865   | 612,705   |
> | ~          | x2    | 88,122,881  | 1,078,225 | 686,225   | 1,083,105 |
> | ~          | x4    | 173,241,857 | 2,019,025 | 989,585   | 2,023,905 |

---

### Author Response · Authors · 2022-11-18
**Summary of Revising**

Thank all the reviewers for the insightful reviews and valuable feedback!
As suggested in the review comments, we made minor revisions to the paper, which we summarize here. Modifications are marked in blue in the new paper for quick reference.

1 As mentioned by reviewer 1, in the proof of theorem 1, the coefficient $2\sqrt{m}$ could be tightened to $\sqrt{2m}$, which appears when we deal with $C_1-C_2$.

2 As suggested by reviewer 2, we revised Subsection 4.1 to correct a few errors, and make it easier to read. We also add a notation that the substructures we considered are all induced graphs in preliminary.

3 As suggested by reviewers 1 and 2, we add empirical study about our theoretical results. Due to page limitation, we add these results in Appendix.

4 As an explanation of the motivation for the model questioned by reviewer 3, we have added a short paragraph after Theorem 2 to further illustrate the motivation for introducing substructures.

5 As suggested by reviewer 4, we discussed model time complexity. We also add ablation study about neighbor size. Due to page limitation, we add these results in Appendix.

---

### Decision · Program_Chairs · 2023-01-20

**Decision:**

Accept: poster

**Justification For Why Not Higher Score:**

There are still some issues, particularly on the meaningfulness of the theoretical results, as well as writing quality and clarity.

**Justification For Why Not Lower Score:**

The paper presents an interesting and well-motivated analysis on the issue of depth in graph transformers, as well as interesting empirical and theoretical findings that can spur further analysis and investigation.

**Metareview: Summary, Strengths And Weaknesses:**

The paper proposes the "DeepGraph" approach which aims to alleviate the limitations in deepening graph transformers. They argue that capturing substructure information becomes increasingly harder as the depth of the model increases, and hence proposes a variant of graph transformers which incorporates substructure tokens and attention into the transformer model.

In general, reviewers appreciated the conceptually simple idea, and agreed that the approach is well-motivated by a clear goal of deepening graph transformers, including theoretical analysis on attention capacity, and the empirical gains are overall convincing. At the same time, some issues were also raised, particularly regarding the meaningfulness of the theoretical results, and some clarity issues.

During the rebuttal stage, the author response made a range of improvements to the paper, which helped to address some of the reviewer concerns and improve the paper, such as adding improvements to the clarity, and adding various empirical studies regarding time complexity, ablations, empirical study of theoretical results, etc. I thank the authors for their efforts in addressing the reviewer concerns and making comprehensive improvements to the paper, which has significantly improved its quality.

Since this was a borderline paper, a virtual meeting was conducted. Reviewers generally agreed that many of the initial concerns had been successfully addressed. There were some questions regarding the relationship of the method to related work, such as TokenGT, but we feel that these were well-addressed by the authors. Overall, reviewers agreed that while the model has similarity to some approaches such as subgraph GNNs, such subgraph approaches are less well explored for Transforrmer models, so there is still sufficient innovation compared to existing approaches. With regard to writing quality, reviewers felt that the paper has been improved significantly and is of satisfactory quality, though some typos / editing may still be required.

There were still some remaining questions, particularly regarding the meaningfulness of the theoretical results; however, on the whole we feel that the paper still presents an intriguing angle on the issue of depth in graph transformers, as well as empirical findings which can spur follow-up investigations on this question, by providing a clear analysis on the depth limitation issue, showing that their model successfully alleviates this issue, and presenting theoretical analysis on the attention capacity issue.

In the end, reviewers and AC find that the paper is well-motivated, and provides a novel and interesting angle on the depth limitation of graph transformers, as well as theoretical analysis and convincing performance gains. Hence, I recommend acceptance.

**Note From Pc:**

if the above contains the word "oral" or "spotlight" please see: "oral" presentation means -> notable-top-5% and "spotlight" means -> notable-top-25%. As stated in our emails, we are disassociating presentation type from AC recommendations

**Summary Of Ac-Reviewer Meeting:**

Reviewers agreed that many of the initial concerns had been successfully addressed, e.g. the relationship of the method to related work  which were well-addressed by the authors. Reviewers also found the writing quality to be significantly improved and sufficient. There were some remaining questions on the meaningfulness of the theoretical results; but reviewers agreed that overall, the paper still presents an intriguing and well-motivated analysis on the issue of depth in graph transformers, as well as interesting empirical and theoretical findings that can spur further analysis and investigation.